# Non-synaptic interactions between olfactory receptor neurons, a possible key feature of odor processing in flies

**Mario Pannunzi**◉*, **Thomas Nowotny**◉

School of Engineering and Informatics, University of Sussex, Brighton, United Kingdom

* mario.pannunzi@gmail.com

## Abstract

When flies explore their environment, they encounter odors in complex, highly intermittent plumes. To navigate a plume and, for example, find food, they must solve several challenges, including reliably identifying mixtures of odorants and their intensities, and discriminating odorant mixtures emanating from a single source from odorants emitted from separate sources and just mixing in the air. Lateral inhibition in the antennal lobe is commonly understood to help solving these challenges. With a computational model of the *Drosophila* olfactory system, we analyze the utility of an alternative mechanism for solving them: Non-synaptic ("ephaptic") interactions (NSIs) between olfactory receptor neurons that are stereotypically co-housed in the same sensilla.

We find that NSIs improve mixture ratio detection and plume structure sensing and do so more efficiently than the traditionally considered mechanism of lateral inhibition in the antennal lobe. The best performance is achieved when both mechanisms work in synergy. However, we also found that NSIs *decrease* the dynamic range of co-housed ORNs, especially when they have similar sensitivity to an odorant. These results shed light, from a functional perspective, on the role of NSIs, which are normally avoided between neurons, for instance by myelination.

**Data Availability Statement:** Data and code are available at: https://github.com/mariopan/flynose.

**Funding:** MP and TN received funding from the Human Frontiers Science Program, Grant RGP0053/2015 (odor objects project) - TN was the

## Author summary

Myelin is important to isolate neurons and avoid disruptive electrical interference between them; it can be found in almost any neural assembly. However, there are a few exceptions to this rule and it remains unclear why. One particularly interesting case is the electrical interaction between olfactory sensory neurons co-housed in the sensilla of insects. Here, we investigate a computational model of the early stages of the *Drosophila* olfactory system and observe that the electrical interference between olfactory receptor neurons can be a useful trait that can help flies, and other insects, to navigate the complex odor plumes in their natural environment.

Our modelling results shed new light on the trade-off of adopting this mechanism: We find that the non-synaptic interactions (NSIs) improve both the identification of the

grant recipients, the European Union's Horizon 2020 research and innovation program under Grant Agreement No. 785907 (HBP SGA2) and a Leverhulme Trust Research Project (RPG-2019-232). MP received a salary from Leverhulme Trust Research Project. The funders had no role in study design, data collection and analysis, decision to publish, or preparation of the manuscript.

**Competing interests:** The authors have declared that no competing interests exist.

concentration ratio in mixtures of odorants and the discrimination of odorant mixtures emanating from a single source from odorants emitted from separate sources—both highly advantageous. However, they also tend to decrease the dynamic range of the olfactory sensory neurons—a clear disadvantage.

## Introduction

Flies, as most other insects, rely primarily on olfaction to find food, mates, and oviposition sites. During these search behaviors, they encounter complex plumes with highly intermittent odor signals: Odor whiffs are infrequent and odor concentration varies largely between whiffs [1–3]. To navigate a plume and successfully achieve their objectives, flies must decipher these complex odor signals which poses several challenges: Identifying odors, whether mono-molecular or a mixture; Identifying odor intensity; Discriminating odorant mixtures emanating from a single source from those emanating from separate sources; identifying source locations, etc. Early sensory processing is understood to play an important role for solving these challenges [4–6]. For instance, lateral inhibition in the antennal lobe is commonly understood to be useful for decorrelating odor signals from co-activated receptor types. Here we investigate the hypothesis that the early interactions between ORNs in the sensilla are similarly, if not more, useful for decoding information in odor plumes.

In both, vertebrates and invertebrates, odors are sensed by an array of numerous receptor neurons, each typically expressing receptors of exactly one of a large family of olfactory receptor (OR) types. In *Drosophila*, olfactory receptor neurons (ORNs) are housed in evaginated sensilla localized on the antennae and maxillary palps [7], each sensillum containing one to four ORNs of different types [7, 8]. The co-location of ORN types within the sensilla is stereotypical, i.e. ORNs of a given type "a" are always co-housed with ORNs of a specific type "b" [8]. Furthermore, ORNs within the same sensillum can interact [9–13] without making synaptic connections (see Fig 1a). While the interactions are sometimes called "ephaptic", referring to their possible electric nature [14], we here prefer to call them non-synaptic interactions (NSIs), for the sake of generality. Whether stereotypical co-location of—and NSIs between—ORNs have functions in olfactory processing and what these functions might be remains unknown, even though several non-exclusive hypotheses have been formulated (see Discussion and [6, 11, 14] and references therein).

Here, we investigate three hypotheses: First, NSIs could help the olfactory system to identify ratios of odorant concentrations in mixtures more faithfully by enhancing the dynamic range over which ORN responses contain the relevant information to do so (see Fig 1b). Second, NSIs could help improve the spatiotemporal resolution of odor recognition in complex plumes (see Fig 1c). Third, NSIs could improve the dynamic range of ORNs' responses to individual odorants by partially removing the ceiling effect that occurs for high concentrations (see Fig 1d). For the first two hypotheses, the NSI mechanism has to compete with lateral inhibition in the antennal lobe, which is commonly recognized to fulfill these roles, even though, of course, the two mechanisms are not mutually exclusive.

Indirect support for the first hypothesis is found in the context of moths' pheromone communication. In some moth species, pheromone mixture ratio discrimination is critical for survival and therefore even slight changes in pheromone component ratios of 1–3% can cause significant changes in behavior. In these species, the ORNs responding to pheromone components are more likely to be co-housed. Meanwhile, when mixture ratios are not as critical for behavior, i.e., significant changes in behavior only occur if pheromone component ratios

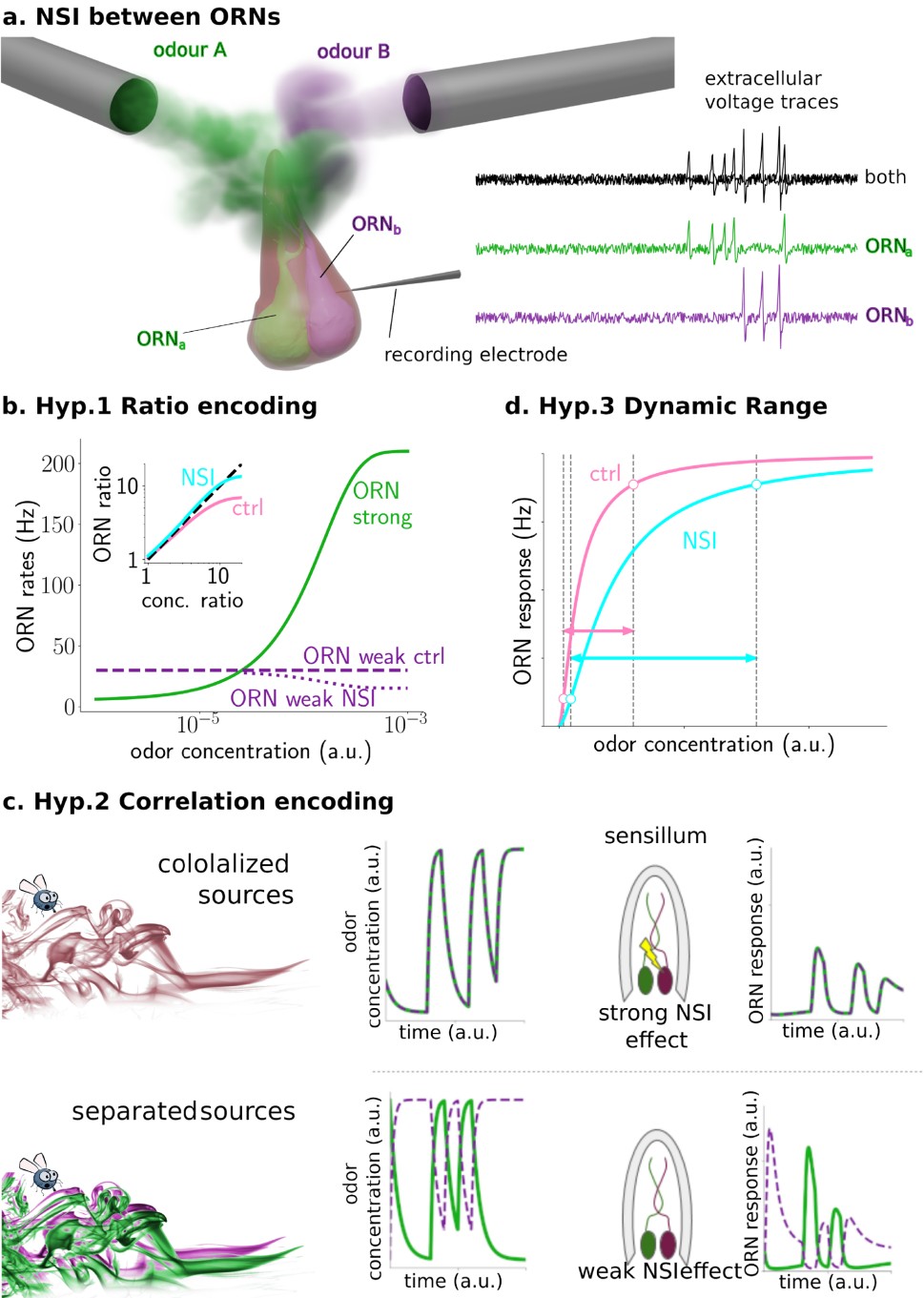

**Fig 1.** a) **Non-synaptic interaction (NSI)**. Theoretical and experimental studies have proposed that the NSI between ORNs is mediated by a direct electrical field interaction between such closely apposed neurons. b) **Hypothesis 1**: Due to the sigmoidal shape of the dose-response curve (green line), the concentration ratio is not well encoded (pink line in inset). The second component's and hence second ORN's activity is assumed constant in this example (purple dashed line). When NSIs are present, the second ORN is inhibited by the first more strongly activated ORN (purple dotted line), which improves ratio encoding by making the encoding ratio (cyan line in the inset) more similar to the concentration ratio (black dashed line). c) **Hypothesis 2**: If a single source emits an odorant mixture (top panel), odorants will arrive in close synchronization, NSIs will take effect and the response in both ORNs is affected. If separate sources emit the odorants (bottom panel), they will arrive in a less correlated way, and NSIs have almost no effect, resulting in larger ORN responses. ORN response data shown is based on a preliminary model. d) **Hypothesis 3**: The lateral inhibition caused by the presence of NSIs decrease the ceiling effect for high concentrations and therefore shift the right boundary of the dynamic range to higher concentrations, so increasing the dynamic range.

change 10% or more, ORNs are less likely to be paired in the same sensilla (see [11] and references therein). The problem when encoding ratios of odor concentration with activity of PNs (or ORNs) is that, while concentration spans several orders of magnitude, the neuron firing rates can only span 2 orders of magnitude. Stimulus intensities in receptor neurons are commonly considered to be coded by spiking frequency (at least since [15]). A concentration ratio around 10 can be difficult, or impossible, to encode if the weakly activated ORN's activity is already around 50 Hz. NSIs would decrease the activity of the weakly activated ORN, inhibited by the activity of the more strongly activated ORN. In this way, the encoding of the ratio can be improved (see Fig 1b).

The improvement of spatiotemporal resolution in the second hypothesis can be achieved by decorrelating odor response profiles to improve odor recognition (see Fig 1c), again much like lateral inhibition in the antennal lobe (AL), or centre-surround inhibition in the retina. Odorants dissipate in the environment in complex, turbulent plumes of thin filaments of a wide range of concentrations, intermixed with clean air. Odorants emanating from the same source presumably travel together in the same filaments while odorants from separate sources are in separate strands (see, e.g., [16] for empirical evidence for this intuitive idea). Insects are able to resolve odorants in a blend and recognize whether odorants are present in a plume and whether or not they belong to the same filaments [17–21]. In the pheromone sub-system of moths, it is known that animals are able to detect, based on fine plume structure, whether multiple odorants have been emitted from the same source or not [17, 18, 22]. In the pheromone subsystem of *Drosophila*, ORNs responding to chemicals emitted by virgin females and ORNs responding to chemicals emitted by mated females are co-housed in the same sensilla: The 'virgin females ORNs' on the males' antenna promote approach behavior, but the 'mated females ORNs' inhibit 'virgin females ORNs' [23]. This inhibition could be implemented through NSIs [8, 11, 23, 24].

The third hypothesis assumes that NSIs can affect the dynamic range and the odor detection threshold of the affected ORNs. It was motivated by the work of Vermeulen and Rospars [14] who first modelled the ORNs' interactions with an electrical circuit, hypothesizing that electrical insulation of the sensillar lymph from the hemolymph generates a common potential difference between the dendritic and axonal compartments of the ORNs within a sensillum. As a consequence, when an odorant activates an ORN, the resistance of the insulation would drop and the activity of the co-localized ORN would show a reduced receptor potential. At the same time, the receptor potential of the activated neuron would be increased, leading to a stronger response of the more sensitive ORN and a slightly lower odor detection threshold. These effects are strongest if the sensitivities to the analyzed odorant are similar. The model has been referred to as evidence for the benefits of co-housed ORNs with NSIs throughout the literature [8, 10, 13, 25, 26]. Here, we analyze the effects of NSIs for an individual odorant in more detail beyond the previously considered steady state equations and with inclusion of the known spike rate adaptation in ORNs (see e.g. [27, 28]).

The experimental evidence for these hypotheses and for the general relevance of NSIs for olfactory processing remains mixed and research is still at an early stage (but see [6] for a more detailed analysis). Encouraged by the available evidence, and without trying to rule out other hypotheses (for further analysis see Discussion), our goal is to investigate, with a computational model, the viability of the hypothesized functions of NSIs between ORNs. Our computational approach helps experimental studies to refine hypotheses about NSIs and eventually answer the pertinent question why such a mechanism that appears to duplicate what is already known to be implemented by local neurons in the AL could nevertheless provide an evolutionary advantage.

A number of computational models have been developed to capture different aspects of the olfactory system of insects. However, until recently, most modeling efforts were based on the assumption of continuous constant stimuli, which are partially realistic only for non-turbulent fluid dynamics regimes (see [29], and references therein). Most commonly, insects encounter turbulent regimes, in which odorant concentration fluctuates rapidly (see S5 Fig).

To cope with these more realistic stimuli, [27, 30–32] have formulated new models of *Drosophila* ORNs, that are constrained by experimental data obtained with more rich, dynamic odor inputs, including a model simulating ORNs and PNs that are subject to input from simulated plumes [32] with statistical properties akin to those of naturalistic plumes (see more details in Model and methods and Correlation detection in longer, more realistic plumes).

Here, we present a network model with two groups of ORNs, each tuned to a specific set of odorants, connected to their corresponding glomeruli, formed by lateral neurons (LNs) and PNs, following the path started by [33, 34], and subsequently by [35–37]. We model the ORNs in a similar approach as [27, 30] with minor differences in the filter properties and the adaptation (see Model and methods). We have tested the behavior of this network in response to simple reductionist stimuli (as commonly used in the literature, see above), and simulated naturalistic mixtures plumes (as described by the experiments in [2, 3]). We then used this simple but well-supported model to investigate the role of NSIs for odor mixture recognition.

## Results

To investigate the role of NSIs in olfactory sensilla, we have built a computational model of the first two processing stages of the *Drosophila* olfactory system. In the first stage, ORN responses are described by an odor transduction process and a spike generator (see Model and methods), in line with previous experimental and theoretical studies [27, 28, 30, 38]. We simulated pairs of ORNs expressing different olfactory receptor (OR) types, as they are co-housed in sensilla. NSIs between co-housed pairs effectively lead to their mutual inhibition (see green trace in Fig 1a). The second stage of olfactory processing occurs in the AL, in which PNs receive input from ORNs and form local circuits through LNs. ORNs of the same type all make excitatory synapses onto the same associated PNs. PNs excite LNs which then inhibit PNs of other glomeruli but not the PNs in the same glomerulus (see Fig 2 and Model and methods for further details).

For maximum clarity, we here focus on only one type of sensillum and hence two types of ORNs that we denote as $ORN_a$ and $ORN_b$. We initially assume that odorants labeled A and B selectively activate $ORN_a$ and $ORN_b$, respectively (see Figs 2 and 1A), but we relax this constraint when analyzing the effects on the ORNs' dynamic range. This assumption is not only sensible for a reductionist analysis of the role of NSIs, but it is also based on experimental observations. For instance, pheromone receptors in moths and in *Drosophila* are highly selective, paired in sensilla, and exhibit NSIs [11, 39]. In the general olfactory system of *Drosophila*, neurons ab3A and ab3B in sensillum ab3 are selectively sensitive to 2-heptanone and Methyl hexanoate, and when stimulated simultaneously they inhibit each other through NSIs [10].

### Constraining the ORN model to biophysical evidence

In this investigation we are particularly interested in the complex time course of odorant responses and have therefore focused on replicating realistic temporal dynamics of the response of ORNs at multiple time scales. ORN responses were constrained with experimental data obtained with delta inputs, i.e. inputs of very short duration and very high concentration,

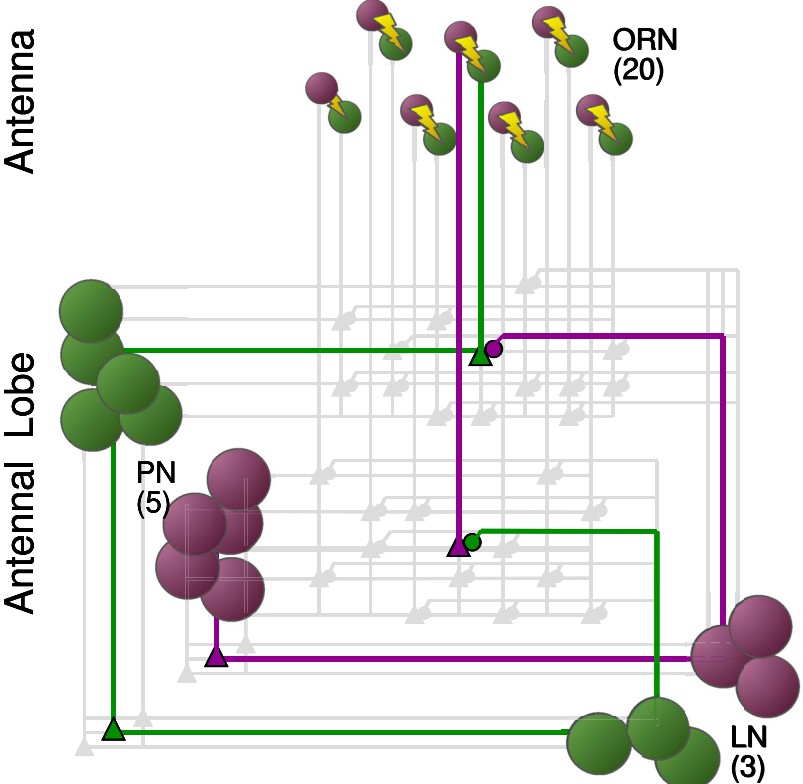

**Fig 2. Model of the early stages of the olfactory system of the *Drosophila*.** The model simulates odor transduction, ORNs and the AL using a minimal subsection of two groups of ORNs ($ORN_a$ and $ORN_b$) and their respective PNs and LNs. We assume that each ORN type, a and b, is tuned to a specific set of odorants (e.g. individual pheromone components) and converges onto its corresponding PNs. PNs impinge onto their respective LNs, and receive inhibitory input from LNs of the other type.

and random pulses, i.e. series of input pulses which durations and inter-stimulus-intervals were drawn from distributions (see S5 Fig) extrapolated from recordings of odorant activity in open spaces [1–3]. We found that our model reproduces the data to a similar quality (percentage error of around 10% for high activity, > 30 Hz) as previous linear-nonlinear models [27, 28, 30, 38, 40], even though it has fewer free parameters (see Fig 3).

To further constrain the model, we compared its results to electrophysiological recordings from ORNs [27, 38] responding to 2 s long odor stimuli with shapes resembling steps, ramps, and parabolas (see S2 Fig). The model reproduces all key properties of the experimentally observed ORN responses. For the step stimuli, ORN activity peaks around 50ms after stimulus onset and the peak amplitude correlates with the odor concentration (see S1a and S1b Fig). After the peak, responses gradually decrease to a plateau. Furthermore, if normalised by the peak value, responses have the same shape independently of the intensity of the stimulus [28], see Fig 3f and 3g. For the ramp stimuli, ORN responses plateau after an initial period of around 200 ms, encoding the steepness of the ramp (see S1c and S1d Fig). More generally, ORN responses seem to encode the rate of change of the stimulus concentration [27, 38, 40]. Accordingly, ORN activity in response to the parabolic stimuli is like a ramp (see S2e and S2f Fig). Parameters used for the transduction and the ORN dynamics are reported in Table 1.

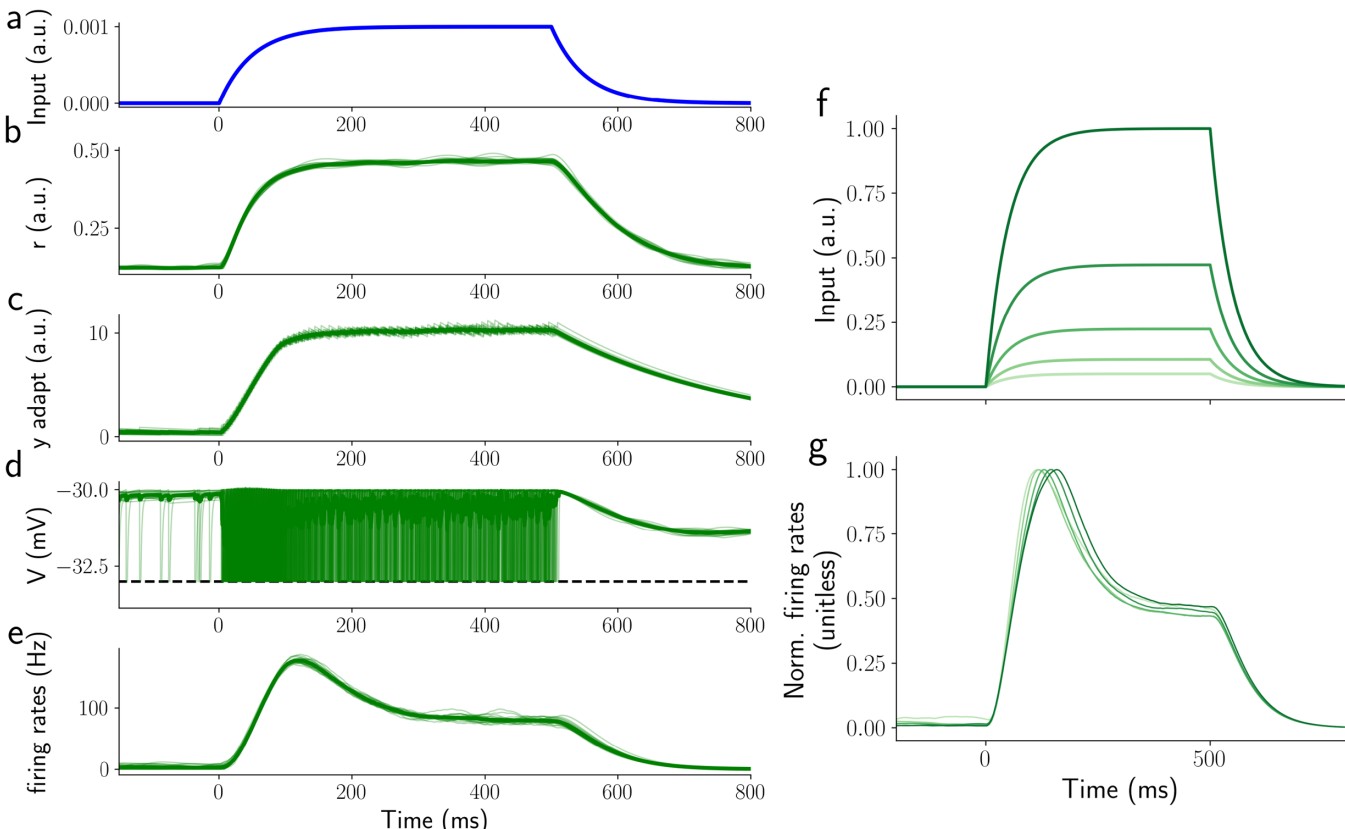

**Fig 3. ORN responses to a 500 ms single step stimulus.** a) stimulus waveform, b) response of the activation variable *r* of the transduction phase, c) activity of the internal ORN variable y (see Model and methods), d) ORN potential response of all ORNs, e) Spike density function of the ORN population activity, f) stimulus waveforms for different odorant concentrations, g) ORN activity normalized to the maximum activity (see Analysis section in Model and methods). Odor concentration is indicated with different shades of green. After normalization, the responses are almost identical to those reported by [28].

Finally, we fitted the parameters of PNs so that the PN responses with a single constant step stimulus resemble the qualitative behavior reported in [41]. We used a brute force search algorithm to find a good starting point and then refined the fit with a Trust Region Reflective algorithm. In that study, the authors reported that the response of PNs to such a stimulus is best described by a sigmoid,

$$v_{PN} = v_{max} \frac{v_{ORN}^{1.5}}{\sigma^{1.5} + v_{ORN}^{1.5}} \tag{1}$$

where $v_{max}$ is the maximum firing rate of ORNs, $\sigma$ is a fitted constant representing the level of ORN input that drives half-maximum response, and $v_{ORN}$ and $v_{PN}$, are the average firing rates of the ORNs and the PN over the stimulation period (500 ms), respectively. Our model is able to reproduce this behavior (see Fig 4e). LNs follow the same behavior (see Fig 4f). Note that this result, i.e. the sigmoidal behavior, generalizes to both, shorter stimulation times (50 and 100 ms, see S3 and S4 Figs) and to the maximum activity (see Analysis section in Model and methods) instead of the time averaged activity. See Table 1 for the remaining parameters of the model.

With a model in place that demonstrates the correct response dynamics for a variety of stimuli, we then analyzed its predictions on whether NSIs can be beneficial for odor processing.

**Table 1. Model parameters.** To reproduce the experimental data in our model, we used the following parameters: Transduction, LIF ORNs, PNs and LNs, and Network parameters. We fitted a subset of ORN, PN and LN parameters in order to reproduce the time course shown in e-phys experiments (e.g. [27, 28]); we obtain similar correlated values as those reported in [84] by fitting the amount of noise injected in the AL ($I_{noise}^{PN/LN}$) and in the transduction layer ($c_0$); Network parameters are not fitted, but extracted from the literature (e.g. [84, 90, 91]). NSI strength ($\omega_{NSI}$) and synaptic strength of LNs ($\alpha_{LN}$) are not fitted, but their values were changed to explore the network behavior. An asterisk indicates the parameters that were not changed during model fitting.

| Transduction | | |
|---|---|---|
| n | 0.82 | |
| $\alpha_r$ | 12.62 | kHz |
| $\beta_r$ | 0.077 | kHz |
| $c_0$ | 1.85e-04 | |
| $r_{noise}$ | 0.5 | |

| LIF ORN | | |
|---|---|---|
| $^*\omega_{NSI}$ | 0.6 | |
| $^*\Theta^{ORN}$ | -30 | mV |
| $^*\tau_{ref}$ | 2 | ms |
| $V_{rest}^{ORN}$ | -33 | mV |
| $V_{rev}^{ORN}$ | 0 | μS |
| $g_l^{ORN}$ | 0.442 | μS |
| $g_r$ | 0.381 | μS |
| $g_y$ | 0.257 | μS |
| $\alpha_y$ | 0.45 | kHz |
| $\beta_y$ | 0.0035 | kHz |

| Network | | |
|---|---|---|
| $^*N_{ORN}$ | 20 | |
| $^*N_{PN}$ | 5 | |
| $^*N_{LN}$ | 3 | |
| $^*N_{glo}$ | 2 | |

| LN and PN | | |
|---|---|---|
| $^*\tau_{ref}$ | 2 | ms |
| $g_l^{LN}$ | 6.2 | μS |
| $g_l^{PN}$ | 10 | μS |
| $^*\Theta$ | -35 | mV |
| $^*V_{rest}$ | -65 | mV |

| LNs | | |
|---|---|---|
| $I_{noise}^{LN}$ | 12 | mV |
| $\alpha_{PN}$ | 0.25 | kHz |
| $\tau_{PN}$ | 19 | ms |
| $g_{PN}$ | 2.1 | μS |

| PNs | | |
|---|---|---|
| $^*V_{rev}^I$ | -80 | mV |
| $^*V_{rev}^E$ | 0 | mV |
| $I_{noise}^{PN}$ | 11 | nA |
| $^*\alpha_{LN}$ | 0.6 | kHz |
| $\tau_{LN}$ | 250 | ms |
| $g_{LN}$ | 0.1 | μS |
| $\alpha_{ORN}$ | 0.5 | kHz |
| $\tau_{ORN}$ | 26.8 | ms |
| $g_{ORN}$ | 0.6 | μS |
| $\alpha_{ad}$ | .02 | kHz |
| $\tau_{ad}$ | 258 | ms |
| $g_{ad}$ | 12.2 | μS |

In particular, we tested the following three hypotheses: 1. NSIs improve the encoding of concentration ratio in an odorant mixture (see section NSIs help encoding odorant concentration ratio), 2. NSIs support differentiating mixture plumes from multiple versus single source scenarios (see section Correlation detection in longer, more realistic plumes), 3. NSIs increase ORNs' dynamic range (see section NSIs alter ORN dynamic range and sensitivity). For the first two hypotheses, we compared four models: a "control model" where the pathways for different OR types do not interact at all, a model with NSIs, but without LN interactions ("NSI model"), a model with lateral inhibition between PNs mediated by LNs in the AL but without NSIs ("LN model"), and a model with both LN interactions and NSIs ("mix model"), see Fig 5. In this way, we are able to test a hierarchy of questions: Are NSIs helpful when encoding a variable of interest (i.e. concentration correlation or ratio) with respect to the control model? How do the NSI and LN models compare when encoding a variable of interest? And, even with the presence of LNs, are NSIs still helpful when encoding a variable of interest?

## NSIs help encoding odorant concentration ratio

Airborne odors travel in complex plumes characterized by highly intermittent whiffs and highly variable odor concentration [1–3]. To successfully navigate such plumes and find for example food, flies must recognize relevant whiffs regardless of the overall odor concentration in them, i.e. perform concentration-independent odorant ratio recognition. This is a difficult problem mainly because PN responses are sigmoidal with respect to concentration. To investigate this problem and understand whether NSIs may play a role in solving it, we stimulated the model ORNs with binary mixtures, varying the overall concentration of the mixture, the concentration ratio, and the onset time of the two odorants. The whiffs in plumes (see e.g. S5b Fig) are mimicked with simple triangular odorant concentration profiles that have a

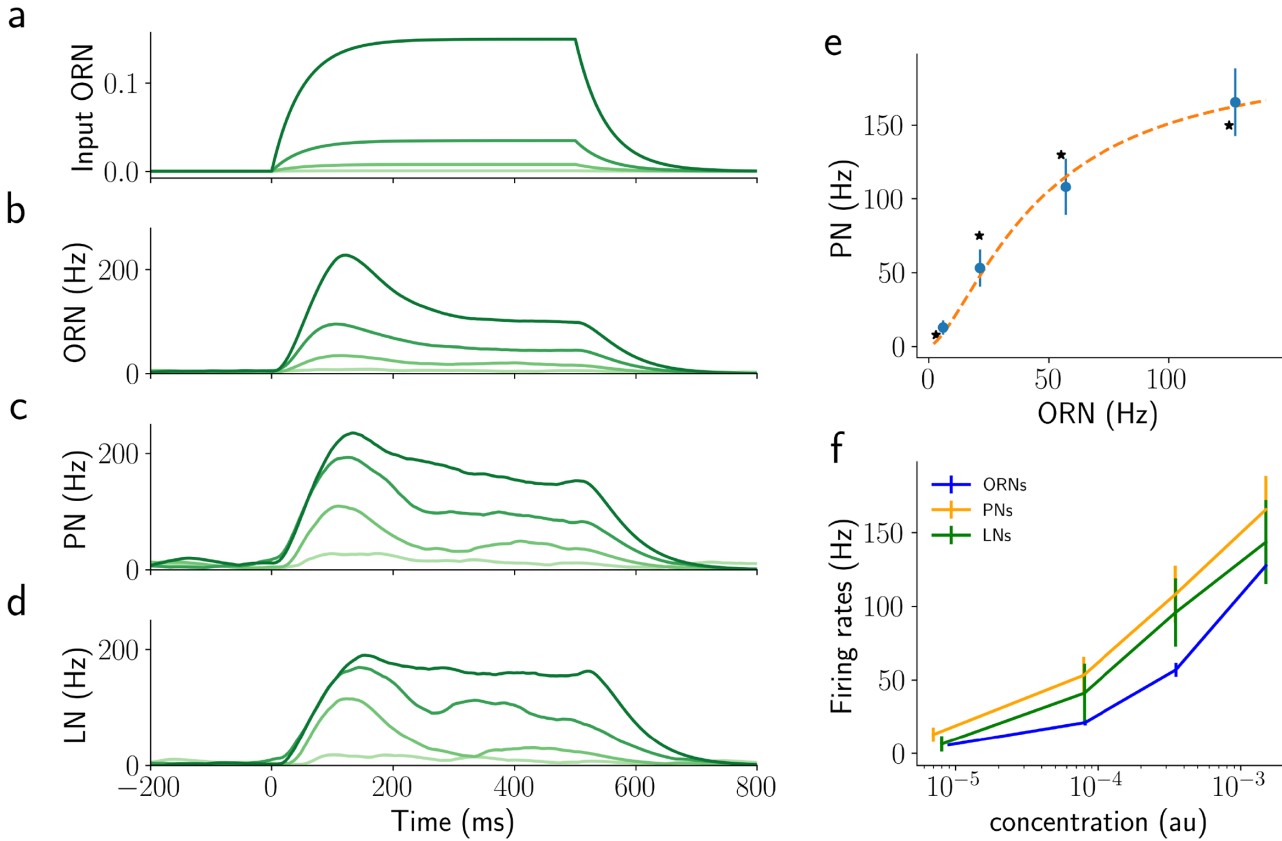

**Fig 4. Network response to 500 ms step stimuli of a single odorant for the network as shown in Fig 2.** a) step stimuli, shades of green indicate concentration, b-d) corresponding activity of ORNs, PNs, and LNs. Shades of green match the input concentrations. e) Average response of PNs over 500 ms against the average activity of the corresponding ORNs. The orange dashed line is the fit of the simulated data using equation Eq 1 as reported in [41]. Asterisks are the values reported in [41]. f) Average values for PNs, ORNs, and LNs for different values of concentration. Note the logarithmic scale of the x-axis in this panel. Error bars show the SEM over PNs.

symmetric linear increase and decrease (see Fig 6). We first analyzed the case of synchronous pulses, which is typical when a single source emits both odorants (see S10a Fig, reproduced from [16]).

Fig 6 shows the typical effects of the NSI mechanism and AL lateral inhibition on PN responses. For the purpose of this figure we adjusted the NSI strength and LN synaptic conductance in such a way that the average PN responses to a synchronous mixture pulse were similar across the LN and NSI models (see Table 1). While the stimulus lasts only 50 ms, the effect on ORNs, PNs and LNs lasts more than twice as long. We observed the same behavior for other stimulus durations (tested from 10 to 500 ms). In the control model (pink), PN responses are unaffected by lateral interactions between OR-type specific pathways and because we have matched the sensory response strength of the two odorants and OR types for simplicity, the responses of the PNs in the two glomeruli are very much the same. For the LN model (orange) the response of ORNs is unaltered by network effects and synaptic inhibition of LNs is the only lateral interaction between pathways. For the NSI model (cyan), ORN activity is directly affected by NSIs and the activity of PNs is lower than in control conditions as a consequence of the lower ORN responses. For the mix model (green), both ORN and PN activities are affected and the PNs' responses are the lowest of the four models. As explained above, NSI strength and synaptic conductance of LN inhibition were chosen in this example so that

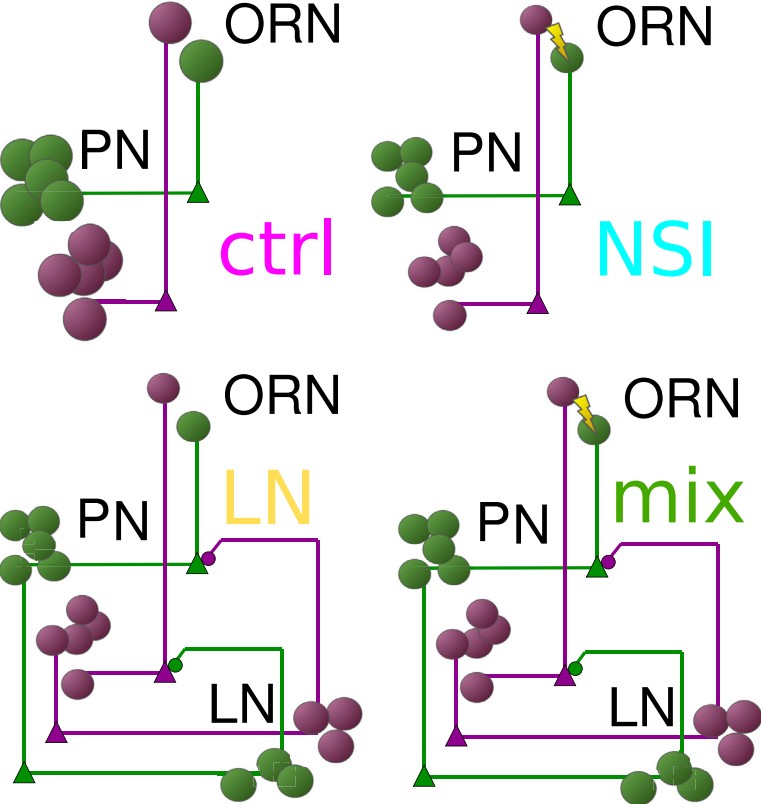

**Fig 5. Schematics of the four models analyzed.** Clockwise from top-left: the *control model* where the pathways for different OR types do not interact at all, the *LN model* with lateral inhibition between PNs mediated by LNs, the *NSI model* without LN interactions, and the *mix model* containing both NSIs and LN inhibition.

the response of the PNs for both models is of similar magnitude. As explained before, the relations between ORN and PN response is non-linear even for the control model.

To investigate the effectiveness of the two mechanisms for ensuring faithful odorant ratio encoding more systematically, we tested the four models with synchronous triangular odor pulses of different overall concentration, different concentration ratios, and for different values of stimulus duration (from 10 to 200 ms), which we selected to match the range of common whiff durations observed experimentally (see S5 Fig). Here, and throughout the study we explored several values from (0.1 to 0.6 for the two strength parameters $\omega_{\text{NSI}}$ and $\alpha_{LN}$). The results are summarised in Fig 7 based on the ratio of maximum activity $R = \nu_b/\nu_a$ (both for ORNs and PNs, see Analysis in Model and methods) during the first 200 ms after the stimulus onset. The values of ORNs and PNs response (measured as maximum activity) is reported in S6 and S7 Figs).

Due to the fundamentally sigmoidal relationship between ORN and PN responses and odorant concentration (see Fig 4e and 4f), the encoding of the ratio between two odorants in a mixture is distorted (see the control model response in Fig 7a and 7b). The encoding of odorant mixtures is indeed already disrupted at the level of the ORNs (Fig 7a), not only on the level of PNs (Fig 7b). Once activated, inhibitory interactions between PNs mediated by LNs slightly improve ratio encoding (Fig 7c and 7d), reducing the strong ceiling effect for high concentrations. The NSI mechanism instead clearly changes the ORN responses in both the NSI

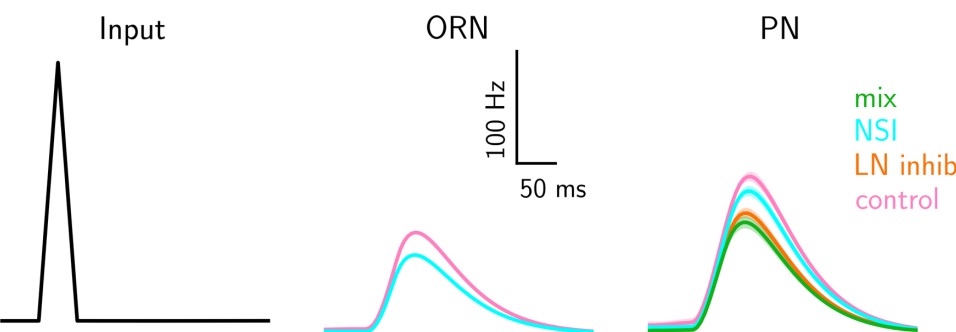

**Fig 6. Responses to synchronous pulses of the control, LN, mix and NSI model.** The time courses of ORN and PN activity are response to a triangular pulse (50 ms, 1st column) for the four models—control (pink), LN model (orange), NSI (cyan), and mix (dark green). The input to the four models is identical, and control and LN models—as well as NSI and mix models—have identical ORN activity, which is therefore displayed only once for each pair. The lines show the average response across repeated runs. The variation across runs is too small to visualise.

model and mix model (Fig 7e–7g), and as a consequence, PN responses change so that their activity reflects the ratio of odor concentrations better for most of the tested concentration ratios (Fig 7f). For the mix model, the LN inhibition generates a further improvement at the level of the PN response and the encoding of this model is better than of all the other models (Fig 7h).

The same results shown for maximum activity are true for the average activity calculated during the first 200 ms after the stimulus onset (see S8 Fig). In the same vein, testing with longer stimuli also yielded qualitatively similar results.

To summarize the encoding quality in each model, we calculated the coding error—the squared relative distance of the PN ratio, $R^{PN}$, and the concentration ratio (see Analysis section in Model and methods). Fig 7i shows the coding error for different values of stimulus duration and concentration averaged over concentration ratios. It is clear that the NSI mechanism boosts the performance for both NSI and mix models as they outperform the LN and control models. This is true for all concentration values tested, but it is less evident for very high concentrations ($\geq 0.005$). While very intuitive, encoding mixture ratios linearly in PN firing rates is not the only option. To analyze encoding quality more generally, we therefore repeated our analysis by calculating the correlation between the odorant concentration ratio and $R^{PN}$. The results from this analysis are qualitatively similar to the results with the coding error (see S9 Fig).

Summarizing the story until now, 1. Both mechanisms help encode odorant concentration ratio, 2. On its own, the NSI mechanism helps encoding it better than the LN mechanism on its own, and 3. the two mechanisms are synergistic so that the mix model, where both mechanisms are present, encodes the concentration ratio best.

In the next section we will explore the effectiveness of the different models for encoding correlations of odorant concentrations in simulated mixture plumes.

## NSIs help encoding correlation of odorant concentration

When odorants are released from separate sources, they form a plume in which the whiffs of different odorants typically are encountered at distinct onset times. To the contrary, when a mixture of odorants is released from a single source it forms a plume where the odorants typically arrive together (see S10 Fig). We hypothesise that if lateral inhibition (via LNs or NSIs)

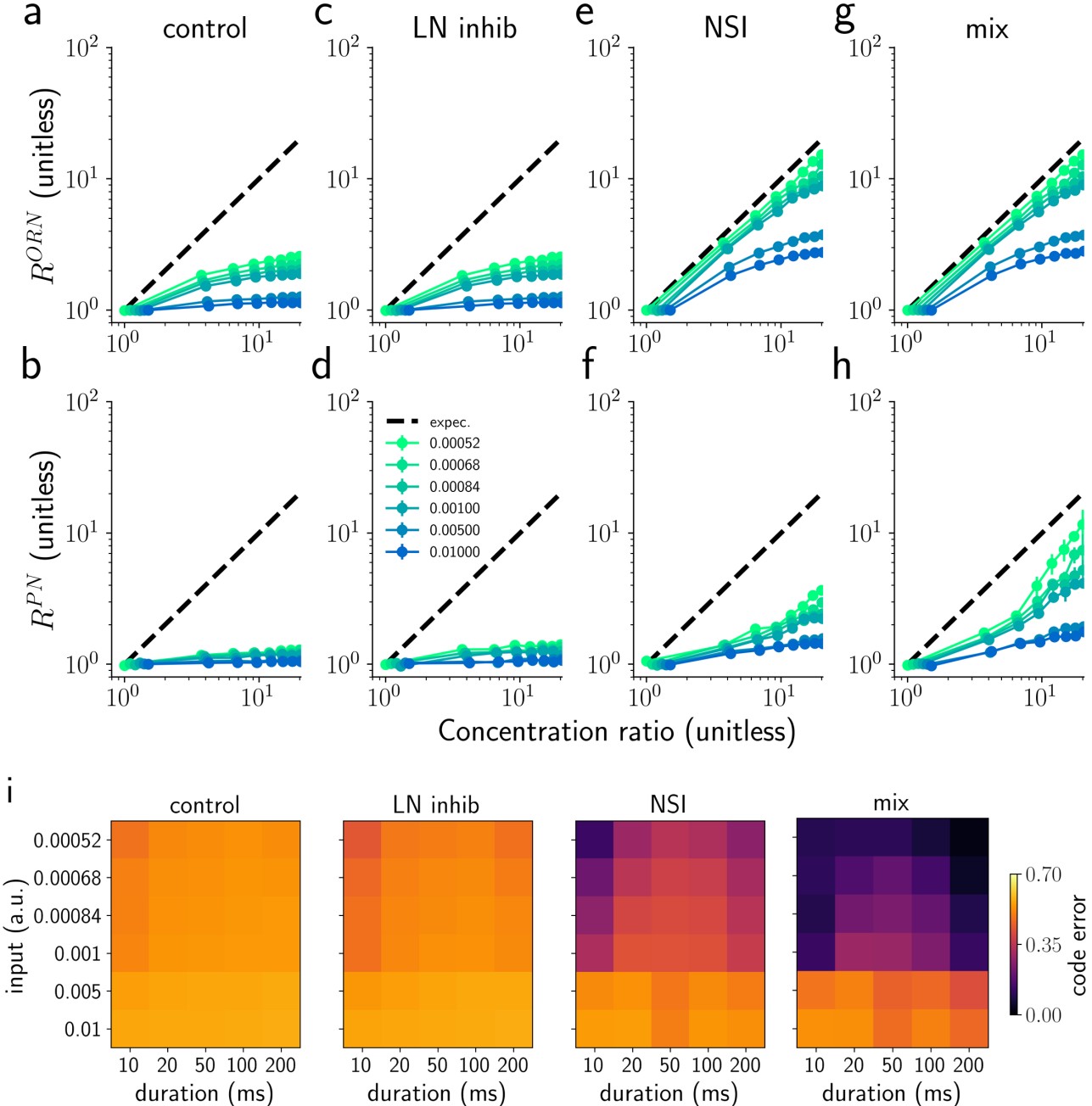

**Fig 7. Encoding concentration ratio with average PN activity.** ORN (a,c,e,g) and PN (b,d,f,h) response ratios ($R^{ORN} = v_b^{ORN}/v_a^{ORN}$ and $R^{PN} = v_b^{PN}/v_a^{PN}$) to a single synchronous triangular pulse of 50 ms duration applied to both ORN groups. The graphs show response ratios versus concentration ratio of the two odorants for six different absolute concentrations of the odorant with lower concentration (colours, see legend in d). The peak PN responses would be a perfect reflection of the odorant concentration if their ratio followed the black dashed diagonal for all concentrations. Error bars represent the semi inter-quartile range calculated over 10 trials. i) Analysis of coding error for different values of stimulus duration and concentration averaged over concentration ratio (see Analysis section in Model and methods). The y axis is the value of the absolute concentration for the odorant with the lower concentration (compare to legend in d).

only takes effect in the synchronous case but not in asynchronous case, it will help distinguishing single source and multi-source plumes. For instance, in the case of pheromone receptor neurons that are co-housed with receptor neurons for an antagonist odorant, the response to the pheromone would be suppressed by NSIs when both odorants arrive in synchrony (same

source) and not when arriving with delays (the pheromone source is separate from the antagonistic source). This is thought to underlie the ability of male moths to identify a compatible female, where the antagonist odorant is a component of the pheromone of a related but incompatible species [18].

To test whether this idea is consistent with the effect of NSIs as described by our model, we first analyzed PN responses to asynchronous pulses (whiffs) of two odorants and then to longer, more realistic plumes. Both analyses were run for our four models—*control*, *LN* model (presenting LN mediated inhibition), *NSI* model (with NSIs but without LN inhibition), and *mix* model (with both NSI and LN inhibition).

**Processing asynchronous pulses.** Before showing the results for the extensive analysis of stimuli with different durations and delays, let us first perform a visual inspection of the four models' responses: Fig 8a shows the responses in the models for the example of two 50 ms triangular odor pulses of the same amplitude and at 50 ms delay. We chose stimuli that excite the two ORN types with the same strength to simplify the analysis and focus on the differences between models with respect to asynchronous input rather than differing input ratios that we analyzed above. In the control model (pink), responses are very similar between $ORN_a$ and

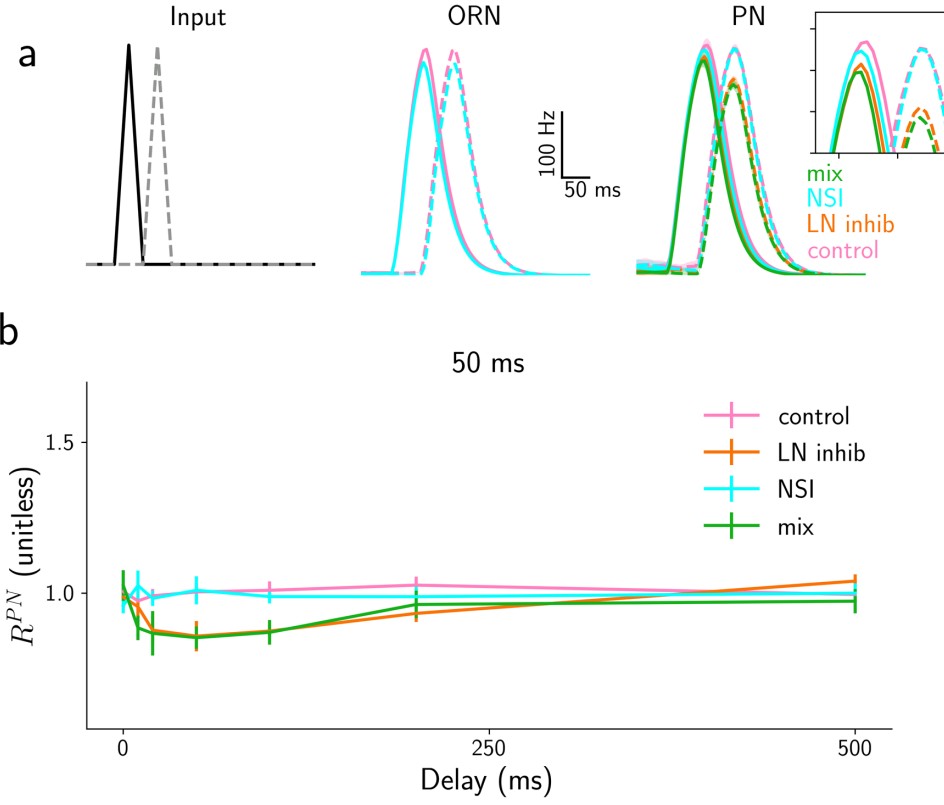

**Fig 8. Responses to asynchronous pulses of the control, LN and NSI model.** a) Time course of ORN and PN activity in response to two asynchronous triangular pulses (duration 50 ms and delay 50 ms) for the four models—control (pink lines), NSI (cyan), mix model (green) and LN model (orange). Input to the four models is identical, while control and LN models and mix and NSI models have identical ORN activity, respectively. These are hence only displayed once for each pair of models. The continuous and dashed lines represent the two odors, ORN and PN types. The lines show the average response across repeated trials. b) Median ratio of the peak PN responses of the two glomeruli $R^{PN} = v_b^{PN}/v_a^{PN}$ in the four models with the same color code for stimulus duration of 50 ms as marked on the top. Error bars represent the semi inter-quartile ranges. For comparison, the same plots are shown for short $\tau_{LN}$ = 25 ms in S12 Fig.

$ORN_b$, as well as, $PN_a$ and $PN_b$ as expected in the absence of interactions and for identical stimulus strength. The NSIs generally decrease the activity of the ORNs in response to both, the first and the second pulse (see cyan lines, green lines for the mix model are not shown as they are identical to the cyan ones). The situation is different for PN responses when LN inhibition is present. Even for the comparatively large delay of 50 ms—the second stimulus (gray dashed line) starts when the first one (black line) ends—for the *LN model* (orange) the $PN_b$ response (dashed orange) is diminished by the inhibition coming through the LNs activated by $PN_a$ (continuous orange). This is a consequence of PN and LN responses outlasting the stimuli as observed above. The situation is identical for the mix model (green lines), while essentially no inhibitory effect is present in the NSI model (cyan lines).

The fact that LN inhibition takes effect for longer time scales than the NSIs is not surprising: While NSIs are (implemented as) instantaneous because they are due to an electric interaction, the LN inhibition is mediated by synaptic activity and therefore follows synaptic time scales, i.e. it depends on the characteristic time of synaptic activation ($\tau_{LN}$). For this figure, we used $\tau_{LN} = 250$ ms for the synaptic decay timescale, in agreement with what has been reported for $GABA_B$ synapses [42, 43]. Hypothesizing a faster time scale instead, with $\tau_{LN} = 25$ ms, the behavior of the *LN model* is, by visual inspection, almost indistinguishable from the *NSI model* (see S12a Fig). However, thanks to an extensive analysis we will see that even for this short decay time constant, the effects of LN inhibition often last long enough to disrupt the second pulse.

To quantify the differences between the four models across different typical conditions, we calculated the ratio between the PN responses of the two glomeruli $R^{PN} = v_b^{PN}/v_a^{PN}$, both for the maximum activity and for the average activity over the stimulation time. Fig 8b shows the results for a single stimulus duration (50 ms) and delays from 0 to 500 ms. In S11 Fig we reported the result for different pulse durations between 10 and 200 ms. The whiff durations and delays were selected to match the range of values commonly observed in plumes experimentally (see S5 Fig).

As expected, the value of $R^{PN}$ is close to 1 for the control model (pink lines) with independent ORNs and PNs and all explored parameters. In contrast, the LN model exhibits clear effects of the lateral interactions in the AL: The response of the second $PN_b$ is suppressed by the response to the first stimulus for all delays shorter than 500 ms. The NSI model shows no suppression, but it is not completely without interference problems: for short delays, when there is an overlap between the two stimuli, and for very high odorant concentrations, the response of the first $ORN_a$ is more inhibited by the activation of the second $ORN_b$, than the second by the first. However, this effect disappears for non-overlapping stimuli. Finally, the mix model (green lines) present exactly the same behavior as the LN model for this analysis, a further indication that NSIs are not relevant for asynchronous stimuli.

The results are very similar whether measured in terms of the maximum activity (Fig 8) or the average activity over the stimulus duration. Even the stimulus duration does not seem to affect the results much (see S11 Fig).

As anticipated, the overall effect on the LN model is diminished for shorter $\tau_{LN} = 25$ ms, see S12 Fig. Therefore, considering faster synaptic interactions, for asynchronous pulses both NSI and LN mechanisms behave substantially like the control model.

**Correlation detection in longer, more realistic plumes.** So far we have seen that NSIs are beneficial for ratio coding in synchronous pulses of odorant mixtures and that they distort responses less than LN inhibition in the case of asynchronous pulses of odorants in mixtures. In this section, we investigate and compare the effects of the two mechanisms when the system is stimulated with more realistic signals of fluctuating concentrations that have statistical features resembling odor plumes in an open field (see S5 Fig). The statistics of the plumes and

how we simulated them are described in detail in the Model and methods; in brief, we replicated the statistical distribution of the duration of whiffs and clean air and the distribution of the odorant concentration which were reported in the literature [2, 3]. Similarly to [32], we simulated plumes as pairs of odorant concentration time series, with a varying degree of correlation to emulate plumes of odors emitted from a single source (high correlation) or from two separate sources (low correlation, see [16] and S10 Fig), while keeping other properties such as intermittency and average odorant concentration constant (see Model and methods and S13 Fig). Similar to the previous section, the stimuli were applied to the models and we analyzed the PN responses in order to understand the ability of the early olfactory system to encode the signal. PN responses are very complex time series (see Fig 9a) and many different decoding

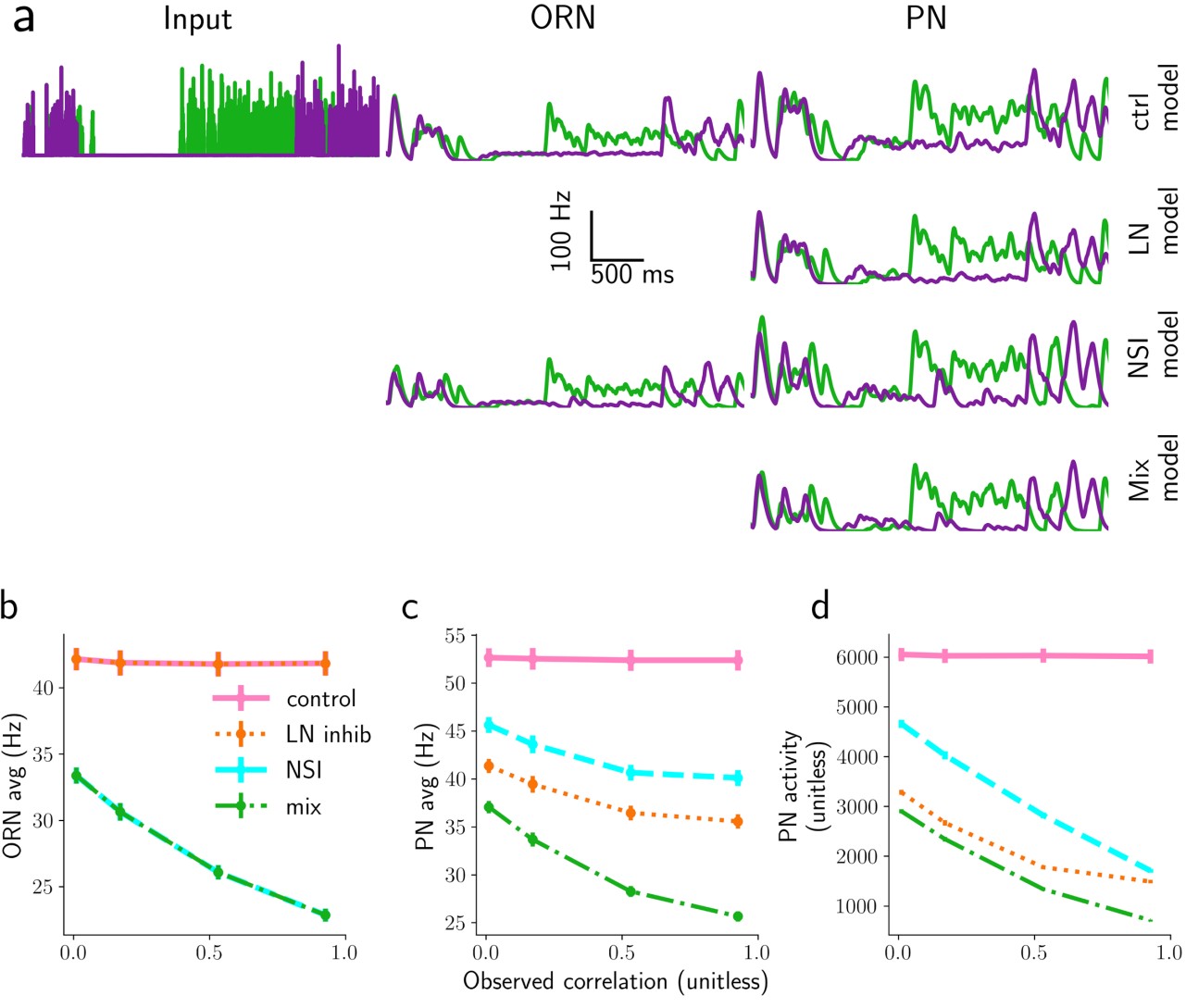

**Fig 9. Responses of the four models to realistic plumes.** a) Time course of stimulus concentration (Input, first column), and response of ORNs (second column), PNs (third column) to two 4 s long simulated plumes with statistical properties as described in the text. Lines are the mean over 10 trials. b) Response of ORNs, and c) response of PNs averaged over neurons and over 200 s for the four models: control model (pink), LN model (orange), mix model (green) and NSI model (cyan). In panel b, the observation from the LN and mix models overlap with those from the control (pink) and the NSI model, respectively. d) PN peak activity above 150 Hz, for 3 s maximum whiff durations.

algorithms [44–48], could be present in higher brain areas to interpret them. However, as a first analysis and like above, we applied the simple measure of peak PN activity in terms of the total firing rate above a given threshold to analyze the quality of the encoding (see Analysis section in Model and methods).

Using this method, we explored plumes with correlation 0 to very close to 1 between the odorant concentration time series and simulating the model for 200 s duration (a few times the maximal timescale in plumes, i.e. 50 s) and inspected the average activity of neuron types over the stimulation period. By construction, the ORN activity for the LN model is the same as the control model (Fig 9a and 9b), while the average ORN activity for the NSI model (cyan) is lower and depends on the correlation between odor signals (Fig 9a and 9b). The mix model (green) has identical ORN activity to the NSI model. These effects are approximately the same for the whole range of the tested NSI strengths $\omega_{NSI}$. The situation is similar for the average PN activity. The average PN response in the NSI, mix and LN models is monotonically dependent on input correlation (see Fig 9c). Hence, both mechanisms are useful for encoding input correlation with the average PN activity. All reported effects remain qualitatively the same for the entire range of explored parameters ($\omega_{NSI}$ and $\alpha_{LN}$).

However, the average activity is probably not highly informative as it does not distinguish between informative responses and simple baseline activity, even during periods of zero odorant concentration (blanks). A very simple more meaningful and easy to implement measure of relevant responses are the average firing rates above a given threshold. This measure is based on the hypothesis that information is conveyed mainly in the high activity, i.e. firing rates above a given threshold, while activity below the threshold is simply baseline noise. In this light we analyzed the "peak PN" response, defined as the integrated PN activity over time windows where the PN firing rate is above a given threshold (e.g. 50, 100, or 150 Hz). Fig 9d shows peak PN for the intermediate 100 Hz threshold (see S14 Fig for plots with other underlying thresholds).

So far we have used simulated plumes with properties corresponding to 60 m distance from the source. At different distances the maximum whiff durations will vary [29]. We therefore asked whether and how the efficiency of the NSI and LN mechanisms depends on maximum whiff duration and hence distance from the source. To address this question, we generated plumes with different maximum whiff duration, $w_{max}$. Fig 10a shows a plot for each tested value of $w_{max}$ (from 0.03 to 50 s) for peak threshold 100 Hz (see S15 and S16 Figs for results with peak thresholds of 50, and 150 Hz). The choice of maximum whiff durations reflects typical experimental observations [2].

Two effects are evident: 1. At zero correlation between the stimuli, PN responses in the NSI model are quite similar to those in the control model while those in the LN and in the mix models differ more, and 2. The PN responses in the NSI and the mix model depend more strongly on the input correlations of the stimuli than the PN activities in the LN model, especially for longer (>3s) whiffs (Fig 10a). This second effect is important because ideally we would like the PN responses to differ maximally between highly correlated plumes and independent plumes in order to discern the two conditions. To quantify these effects we measured the following distances:

- The distance between PN responses of the control model and those of one of the other three models (NSI, LN and mix model) at zero correlation, defined as $p_{ctrl}^0 - p_x^0$ with $x \in$ (NSI, LN and mix model) (Fig 10b)

- the distance between PN responses for the NSI, mix and LN model at 0 correlation and at correlation (very close to) 1, defined as $p_x^0 - p_x^1$ with $x \in$ (NSI, LN, mix) (Fig 10c).

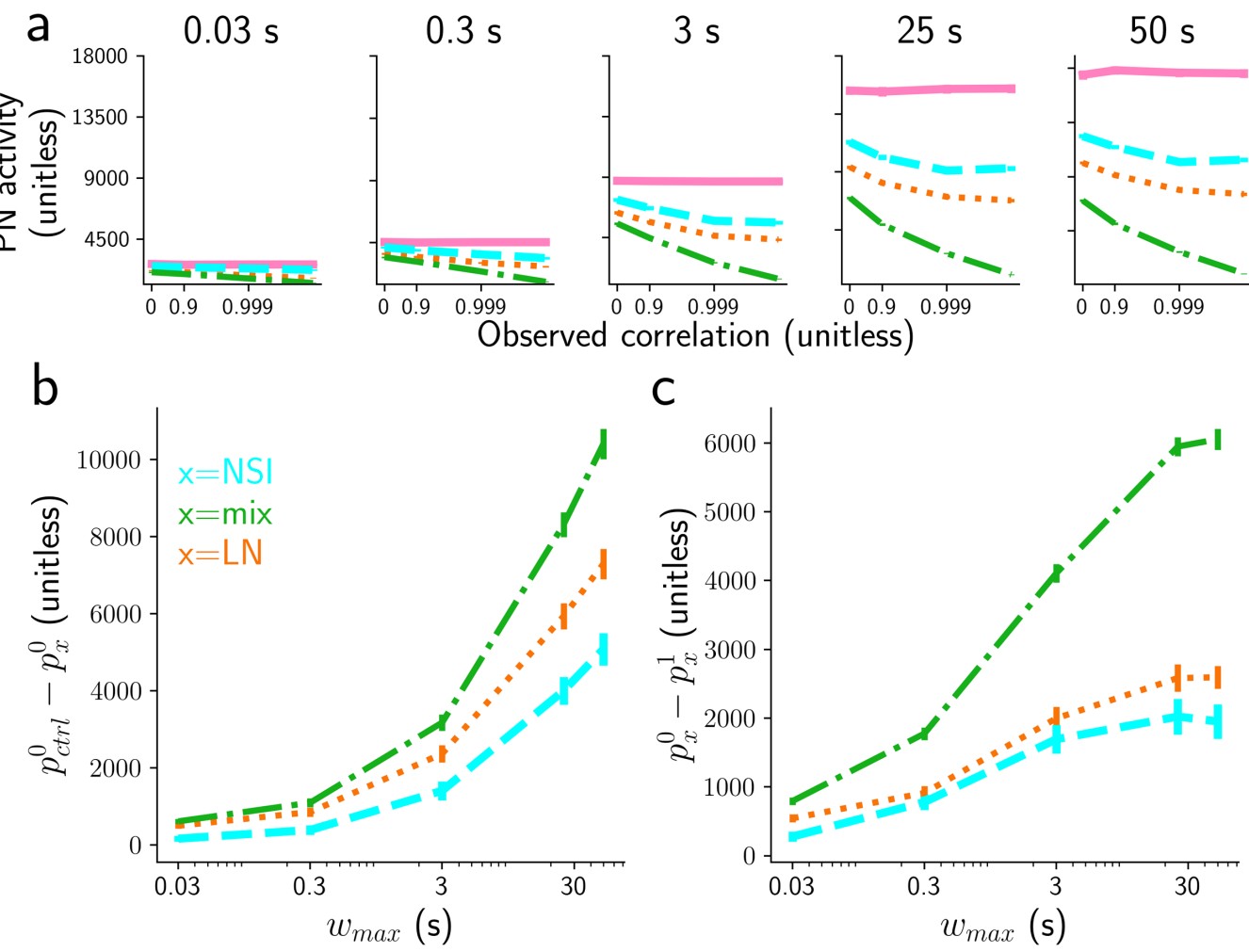

**Fig 10. Comparing encoding efficiency of odorants' correlation.** a) Peak PN response for threshold 100 Hz, and for different subsets of maximal whiff durations (from 0.03 to 50s) for the four models: control model (pink), LN model (orange), mix model (green) and NSI model (cyan). Note that the horizontal axis has a log-scale. b) Distance between the PN activity of the control model and one of the other three models (NSI, LN or mix model), at 0 correlation, $p^0_{ctrl} - p^0_x$ with $x \in$ (NSI, LN, mix). c) Distance between the PN activity at 0 correlation and at correlation 1 for the NSI, LN and mix model, i.e., $p^0_x - p^1_x$ with $x \in$ (NSI, LN, mix).

As expected from previous results on asynchronous pulses, the mix and LN models interfere more at zero correlation than the NSI model and this is true for the three thresholds used (Fig 10b and S15b and S16b Figs). However, when repeating the same measurement with smaller $\tau_{LN}$ = 25 ms, the amounts of interference in the LN and NSI models become identical.

For non-zero correlations, PN responses depend on the plume correlation in the LN, mix and NSI models. Depending on the threshold, NSI or mix models better encode the correlation between odors: for high threshold ($\geq 150Hz$) the NSI model is better, i.e. $p^0_{NSI} - p^1_{NSI}$ is generally higher than $p^0_{LN} - p^1_{LN}$ and $p^0_{mix} - p^1_{mix}$ (S16c Fig); otherwise the mix model outperforms the other models (Fig 10c and S15c Fig). Simulating the LN model with shorter inhibition time-scale ($\tau_{LN}$ = 25 ms), the encoding performances of the LN and NSI models are almost identical.

These last two results demonstrate that the NSI mechanism can be used for encoding correlations between odor plumes, and that for this task its performance is equal to (for short

$\tau_{LN}$ = 25 ms) or better (for larger $\tau_{LN}$) than the performance of the LN mechanism. The mix model exhibits both higher encoding performance and higher interference.

In the next, final section we will analyze how NSIs affect the dynamic range and detection threshold of the ORNs.

## NSIs alter ORN dynamic range and sensitivity

Until now, we have analyzed the effects of NSIs on two neurons and two odorants where each neuron was specifically sensitive to one of the odorants. We now generalize by taking into account the possibility that a single odorant can generate a response in both ORNs. In a theoretical study [14], Vermeulen and Rospars proposed that, in this situation, NSIs may be useful to alter the ORN's dose-response curve. We tested this hypothesis by measuring the odor detection threshold and the total dynamic range of a pair of ORNs co-housed in a single sensillum and stimulated by a single odorant.

We used as stimuli short triangular pulses (with duration of 10, 20, 50, 100 and 200 ms). Dynamic range is measured using the common definition of the logarithmic distance between a lower threshold and a higher threshold. The lower threshold $C_l$ is defined as the concentration at which 10% of the maximum ORN response is first reached and the higher threshold $C_h$ where 90% is first reached. ORN response here is calculated as maximum activity (see Analysis in Model and methods).

Before analyzing the overall effect of the NSIs, let us focus on how the two dose-response curves are changed by them. Fig 11 shows the dose-response curves for ORNs of the control model (on the top left) and the NSI model (on the top right) for two neurons which are equally sensitive to the odorant. We then analyzed ORNs whose sensitivities are increasingly different (from top to bottom of Fig 11). The ORNs' sensitivity distance, SD, is measured as the logarithmic ratio between the lower thresholds of the two isolated ORNs (i.e. $\log_{10}(C_l^b/C_l^a)$, where a and b refer to the two ORNs). We can see that for the NSI model, especially when the SD is high, the dose-response curve of the more sensitive ORN exhibits a non-monotonic peak-and-plateau response that degrades intensity encoding. Indeed, the high threshold of that neuron (green) for the NSI model is typically lower than in the control model. The less sensitive neuron (purple) has a higher value of both, the lower and the higher threshold in the NSI model but overall a smaller dynamic range is observed compared to the control model.

In an attempt to furnish a description of the overall effect of the NSIs on the pair of ORNs, we analyzed and compared the dose-response curves for the average response of the two ORNs (see blue lines in Fig 11). We used the average response for simplicity as the sum is the simplest way to represent the dynamics of both neurons together and because it is resembling the measure adopted by Vermeulen and Rospars [14]. The average response exhibits lower values of both thresholds for the NSI model compared to the control model (see S17 Fig) for several values of the sensitivity distance. This indicates an improved detection threshold. However, because the higher threshold is affected even more by the presence of NSIs, especially when the SD is very low or very high (see S17b Fig), the dynamic range is *smaller* in the NSI model compared to the control model (see Fig 11b).

Although our work differs in many aspects from the earlier theory (full temporal description of stimuli and the ORN and NSI implementation) and we took into consideration the combined dynamic range and not simply that of one ORN, our results are not very dissimilar from those reported by Vermeulen et al. [14]: The NSI mechanism slightly decreases the lower threshold for ORNs that respond equally to an odor but also considerably decreases the upper threshold, hence decreasing the dynamic range. For medium and high sensitivity difference, the dynamic range is either broadly unchanged or decreased, respectively (see Fig 11b).

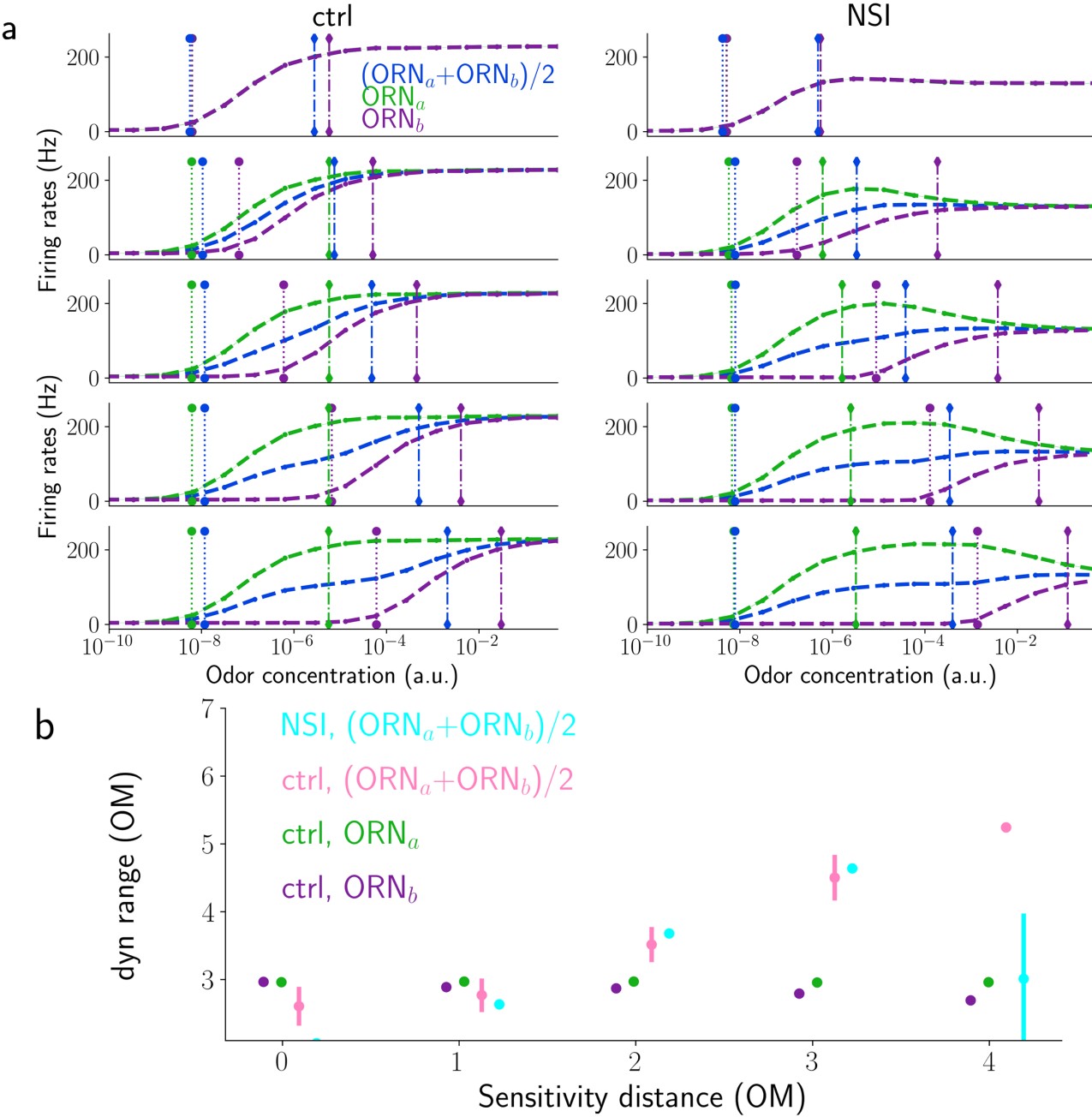

**Fig 11.** a) Dose response curves of the isolated ORNs and of the averaged activity of the two ORNs for the control model (left column) and for the NSI model (right column). Vertical lines indicate the low (dotted) and high (dot-dashed) thresholds. From top to bottom: Three different values of sensitivity distance ($\log_{10}(C_l^b/C_l^a) \approx 0$, 2 and 4). b) Dynamic range of the isolated ORNs (green and purple) and of the two models calculated for several values of the ORNs' sensitivity distance ($\log_{10}(C_b^b/C_l^a)$) in order of magnitude (OM). For the control (pink) and the NSI model (cyan), dynamic ranges are calculated from the average activity of the two ORNs.

## Discussion

*"Thought experiment is in any case a necessary precondition for physical experiment. Every experimenter and inventor must have the planned arrangement in his head before translating it into fact."* E. Mach (1905)

To avoid electric and magnetic interactions between them, neurons are typically myelinated. An exception to this rule are the ORNs of *Drosophila* which interact non-synaptically. While it is clear that this interaction can interfere with the propagation of signals, we have asked whether NSIs could also have a functional role in the olfactory system of insects and what that role could be.

We have implemented a model of the early olfactory system comprising ORNs of two receptor types, their NSIs in the sensillum, and two corresponding glomeruli in the AL, containing PNs and LNs in roughly the numbers that have been observed for *Drosophila*. Our objective was to investigate three potential roles of NSIs in insects' olfactory processing: 1. Concentration invariant mixture ratio recognition, vital for insects to identify the type or state of an odor source (see e.g. [49–53] and references therein); 2. odor source separation, which can be critical for insects, e.g. in the context of finding mates [17, 18, 23]; 3. odor detection and intensity perception [14].

By comparing our model with NSIs to a control model without lateral interactions between pathways for different receptor types, we found evidence that NSIs should be beneficial for concentration invariant mixture processing: NSIs lead to more faithful representations of odor mixtures by PNs in the sense that the ratio of PN activity is closer to the ratio of input concentrations when NSIs are present. Similarly, the correlation between the ratio of input concentrations and the ratio of PN activity is higher for the NSI model. While we admittedly do not know how exactly odor information is represented in PN activity, responses that differ systematically with input ratio must be superior to responses that saturate and hence do not inform about the input ratio, as seen in the control model.

Furthermore, using a model variant with no NSIs but LN inhibition between glomeruli in the AL we found that 1. For synchronous individual whiffs, both models, the one with NSI mechanism and the one with LN inhibition, are better than the control model in several conditions (Fig 7i); moreover, the NSI mechanism is typically more effective than LN inhibition. 2. For the same conditions, the mix model that integrates both NSI and LN inhibition is even more effective at encoding odorant ratio.

These results further support the hypothesis that the NSI mechanism offers an evolutionary advantage by enabling more precise odor coding for these simple stimuli. Similar conclusions can be drawn when analyzing the capacity of the insect olfactory system to encode the correlation between two odorants in a more realistic setting of an odor plume. We found that, when analyzing peak PN activity (the integrated PN firing rate over windows during which it is above a given threshold), the model with NSI mechanism outperforms the LN inhibition model and that both are better when considering peak activity than when considering average PN activity. The better performance of the NSI model was in part anticipated because for asynchronous individual whiffs, PN responses to later whiffs are altered (inhibited) by the response to the earlier whiffs when LN inhibition is present. This effect depends on the time scales of the LNs and the inhibitory synapses. In contrast, with NSIs the PN responses to the second whiffs are only mildly affected (increased) by the activity triggered by the first whiff. This effect is only evident for very high concentration, indicating that with NSIs there is less of a trade-off between benefits in encoding synchronous mixtures and distortions when odorants from separate sources mix.

The third hypothesis tested the idea illustrated in Fig 1d that the ORNs' dynamic range could be improved by NSIs [14, 25, 26]. The improvement of dynamic range by NSIs sits alongside work that showed that syntopic interactions at the receptor level and masking interactions at a cellular level achieve similar effects [54, 55] as well as improving mixture representations. How these receptor-level and cell-level mechanisms interact with sensillum-level NSIs is an interesting future research question.

However, our results indicate that this hypothesis is not supported by the models: The total dynamic range of two isolated ORNs is typically greater than that of two ORNs interacting through NSIs. The disruptive effect of NSIs arises from the dependence of the NSI strength on the activity of the more strongly activated ORN, so that instead of preventing the ceiling effect for high concentrations, NSIs lead to an activity peak for central values of the concentration. The effect increases with the similarity of the sensitivity of the ORNs and in the limit of identical ORNs it is so strong that the resulting dynamic range is inferior to the dynamic range of a single ORN.

Furthermore, the non-monotonic peak-and-plateau response of the more strongly activated ORN degrades intensity encoding. However, the pair of neurons would be able to rapidly detect odorants at specific intensities.

## The model and its limitations

*"A good model should not copy reality, it should help to explain it"*, [56].

As in every modelling work the level of description must match the purpose of the investigation. In terms of Marr's categorisation of models [57], our model is somewhere between the algorithmic level—as both our models implement a form of lateral inhibition—and the implementation level—albeit we are not yet able to capture the details of the underlying physics of the NSI mechanism. Because of our hypothesis that the role of NSIs is to improve processing of temporally complex stimuli, we focused on a description which included temporal dynamics but was otherwise as simple as possible. We therefore have simplified 1. the cellular dynamics of odor transduction [31, 58–62] and only heuristically describe the macroscopic effects at the receptor neuron level, an approach similar to [30]; 2. the complexity of the full receptor repertoire in the insect olfactory system, e.g. about 60 ORN types in *Drosophila*, and instead focused on a single sensillum with two co-housed ORNs; 3. the true complexity of the many different LN types and transmitters in the AL [63], using only GABA-like LNs. 4. the spatial distribution of the sensilla on the surface of the antenna or the maxillary palp; 5. the complexity of odor stimuli delivered by stimulation devices in the experiments we are mimicking for the single pulse investigation (see the corresponding Model and methods section, [29]), 6. the asymmetry of NSIs where there is some evidence that the strength of the NSIs is proportional to the size of the ORN that is exerting the interaction onto another neuron [13]. By making these simplifications we were able to reduce the number of free parameters in the model, reasonably constrain most parameters and scan the few remaining parameters, such as the strength of LN inhibition, across a reasonable range. This increases our confidence that the observed benefits of NSIs for olfactory information processing are not artefacts of particular parameter choices in the model(s).

## The future: How to generalize the model

*"[. . .] I have the solution, but it works only in the case of spherical cows in a vacuum."*,

For the sake of simplicity we chose to work with a specific animal model in mind and because of the large amount of information available in the literature, we chose *Drosophila*. It will be interesting to see whether and how much our results can be generalized to other insects such as other flies or moths (that have 1–4 ORNs per sensillum), or even bees, ants and beetles (with up to 20–30 or even hundreds of co-housed ORNs).

We can only speculate that all the effects of the NSIs shown here will be, at least partially, amplified, as there will be a much higher probability that the ORNs would be concurrently sensitive to the odorants present in odor mixtures. But of course, we have to consider the

disruption generated by the decreased dynamic range whenever co-housed ORNs respond to the same odorants.

These ideas are probably closer to be tested than we thought: For example, ants, belonging to the Hymenoptera order, can have up to two hundreds ORNs in a single sensillum and their interaction has been suggested to be useful to encode membership to the nest [64]. Indeed the authors found that "ant sensilla that are sensitive to a non-nestmate Cuticular Hydrocarbons (CHC) blend also house a large number (about 200) of putative receptor neurons". The idea is that "a large number of receptor neurons might facilitate the resolution of small differences in the multiple-component chemical cues encountered by the sensillum and also might integrate the chemical information as a unit".

However, the complexity of this problem is clearly enough to require an extensive study that we hope can start from this work and the model that is freely available online.

## Comparison with related modelling works

*"If I have seen further it is by standing on the shoulders of Giants."* I. Newton (1675).

Our work builds on ideas in previous models (e.g., [14, 65, 66]) and concurrent approaches (e.g. [30]). While earlier modeling works focused on the oscillatory and patterned dynamics of activity in the antennal lobe [43, 67–71], it was soon realized that the recognition of odorants and their mixtures across different concentrations posed a particularly difficult question. One school of models explored the idea of winnerless competition as a dynamical systems paradigm for concentration invariant coding [72, 73] while others explored more direct gain control mechanisms mediated by local neurons in the AL [35, 36, 74]. The task becomes even more difficult when the exact ratio of mixtures needs to be recognised, and a network model for mixture ratio detection for very selective pheromone receptors has been formulated in [37]. However, generally, odors already interact at the level of individual ORs due to competitive and non-competitive mechanisms which can be recapitulated in models, see e.g. [66] for vertebrates and [31] for invertebrates.

However, our model also makes a clear departure from the large number of models that have been built on assumptions and data based on long, essentially constant, odor step stimuli. While these kind of stimuli are not impossible, they can be considered as the exception more than the rule; for instance, even at more than 60 m from the source, around 90% of whiffs last less than 200 ms [1, 75], see [29] for review. This insight is particularly difficult to reconcile with models that emphasize and depend on intrinsically generated oscillations in the antennal lobe [43, 67–71], and models that depend on comparatively slow, intrinsically generated dynamics such as the models based on the winnerless competition mechanism [72, 76, 77]. The original interpretation of these models, how they use intrinsic neural dynamics to process essentially constant stimuli, is disrupted when stimuli have their own fast dynamics. How to reconcile the idea of intrinsic neural dynamics for information processing with natural odor stimuli that have very rich temporal dynamics of their own remains an open problem.

In building our model, we followed the main ideas developed by [14] but went beyond the assumption of constant stimuli and also added the important element of adaptation in ORNs and PNs, a widely accepted feature that is important in the context of dynamic stimuli. While Vermeulen et al. were already interested in possible evolutionary advantages of NSIs (without finding much), we here added the comparison with lateral inhibition in the AL that has been described as a competing mechanism, from an experimental (e.g. [11]) and a theoretical point of view (e.g., [35–37, 78]. Finally an important addition in this study are the mixture stimuli: Many, though not all, earlier works focused on the response of the network to mono-molecular odors, whereas we analyze the network response to two-odorant mixtures.

A previous study with very similar motivation relating to mixture ratio recognition is the analysis of pheromone ratio recognition of [37]. However, this earlier work still assumed constant stimuli, no adaptation in ORNs, a fixed target input ratio and only LN inhibition.

## Further hypotheses about NSIs

*"There is always a well-known solution to every human problem—neat, plausible, and wrong."*
H. L. Mencken 1920 "Prejudices: Second Series"

At this early stage, our knowledge and understanding of NSIs is still full of gaps. For example, while suggestive, our results cannot prove beyond doubt whether NSIs are effectively useful to the olfactory system, or whether they are just an evolutionary spandrel. We also do not know their evolutionary history. One interesting idea would be that the complex function of improved odor mixture encoding could have arisen as a side effect from a simpler function, e.g. of saving space, but we do not have any evidence to support this.

Researchers in the past 20 years have suggested a number of non exclusive explanations for the functions of NSIs. We have analyzed three of them—improved odor ratio representation, and improved dynamic range, and detecting plume correlations. An alternative hypothesis is that NSIs could facilitate novelty detection for odor signals on the background of other odors [11], if newly arriving "foreground odors" suppress the ongoing response to an already present "background odor". Todd et al. already noticed that NSIs duplicate the role of LNs in the AL even though [7] pointed out later that LN networks take effect later and mainly decorrelate PN activities and normalize them with respect to the *average input* from ORNs. Here we have added to the discussion by showing that NSIs have advantages with respect to their faster timescale that leads to less disruption of asynchronous odor whiffs and that the two mechanisms can synergisticly work together, at least for ratio encoding.

Moreover, NSIs have two additional key advantages with respect to LN inhibition in the AL or processes in later brain areas: 1. NSIs can be energetically advantageous as they don't need spikes [6, 79–83]. 2. NSIs take place at the level of the single sensillum and hence a few spikes and synapses earlier than any AL or later interactions [7, 11]. In the AL the information from ORNs of the same type is likely pooled and information about the activity of individual ORNs is not retained (see e.g. [40, 84]). Therefore, while interactions within the sensillum are precise in space and time, interactions in the AL will be global (averaged over input from many sensilla) and information channels will interact in an averaged fashion. Similar local interactions in the very early stages of sensory perception were already discussed for the retina [85, 86].

## Conclusions

In conclusion, we have demonstrated in a model of the early olfactory system that NSIs could be useful for faithful mixture ratio recognition and plume separation but less so for the dynamic range of ORN responses. On their own they are slightly more efficient than LN inhibition on its own but both can work in synergy. In our future work we seek to confirm the behavioral relevance of NSIs in *Drosophila*. Other interesting future directions include the relationship of NSIs and syntopic mixture effects/masking, as well as the precise differential roles of NSIs and LN inhibition when both are present at the same time.

## Model and methods

### Model topology

We model the electrical activity of the early olfactory system of *Drosophila melanogaster*. The model encompasses ORNs on the antenna, and the matching glomeruli in the AL, containing

PNs and LNs. ORNs are housed in sensilla in pairs, and each neuron in a pair expresses a different OR type. The paired neurons interact through NSIs, effectively leading to mutual inhibition (see Fig 1a). There are multiple sensilla of the same type on each antenna. We here model $N_{ORN}$ = 20 sensilla per type [84]. ORNs of the same type all project exclusively to the same glomerulus in the AL, making excitatory synapses onto the associated PNs.

The PNs are inhibited by the LNs of other glomeruli but not by the LN in the same glomerulus (see Fig 2). The model simulates one type of sensillum and hence two types of ORNs, $ORN_a$ and $ORN_b$. We assume that $ORN_a$ and $ORN_b$ are selectively activated by odorants A and B, respectively (see Figs 2 and 1a).

## Olfactory receptor neurons

We describe ORN activity in terms of an odorant transduction process combined with a biophysical spike generator [30]. During transduction, odorants bind and unbind at olfactory receptors according to simple rate equations. As we are not interested in the competition of different odorants for the same receptors, we simplify the customary two-stage binding and activation model [65, 66, 87] to a single binding rate equation for the fraction r of receptors bound to an odorant,

$$\dot{r} = \alpha_r c^n (1 - r) - \beta_r r \tag{2}$$

The variable $c$ encodes the odorant concentration. The relevant unit is ppm (part per million), even though this number is difficult to obtain and rarely given in the literature for technical reasons (see [29]). Instead, the proxy of concentration in liquid solution v/v is often used and we here use numbers that are compatible with this interpretation. We described the spike generator by a leaky integrate-and-fire neuron with adaptation,

$$C\dot{V} = g_l^{ORN}(V_{rest}^{ORN} - V) + g_y y(V_K - V) + g_r(r + \zeta_r)(V_{rev}^{ORN} - V) \tag{3}$$

$$\dot{y} = \alpha_y \sum_{t_s \in S} \delta(t - t_s) - \beta_y y \tag{4}$$

where $C = 1$ nF is the membrane capacitance, $V_{rest}$ is the resting potential, $V_{rev}$ the reversal potential of the ion channels opened due to receptor activation, $\zeta_r$ is Gaussian colored noise with zero mean and standard deviation $r_{noise}$, representing receptor noise, and $y$ is a spike rate adaptation variable with decay time constant $1/\beta_y$. When $V > \Theta^{ORN}$, the neuron fires a spike and V is reset to $V_{rest}$ and it does not change for a refractory period $\tau_{ref} = 2$ ms.

The parameters $\alpha_y$ and $\beta_y$ are estimated together with $\alpha_r$, $n$ and $\beta_r$ to reproduce the data presented in [28]. We fit with a brute force parameter scan to find a good starting point and then refine the fit with a least squares hill climbing algorithm. The model is similar in nature to the models presented in [30, 88] albeit simplified and formulated in more tangible rate equations. As illustrated in Fig 3 and S1 and S2 Figs, this simplified model can reproduce experimental data equally well as the previous models.

## Non-synaptic interactions

To simulate the NSIs, we assumed that the electric interaction takes place through the reversal potential of the receptor current and that it depends linearly on the receptor activation of the

co-housed neuron. Omitting ORN in the superscript, we obtain the following equations:

$$V_{\text{rev}}^1 = V_{\text{rev}} - \omega_{\text{NSI}} r_2 (V_{\text{rev}} - V_{\text{rest}}), \tag{5}$$

where $0 < \omega_{\text{NSI}} < 1$ is the unitless strength of the NSI.

## The antennal lobe

We here reduce the antennal lobe (AL) to two glomeruli, a and b (see Fig 2) in order to focus on the effects of NSIs of the corresponding ORN types. The numbers of PNs and LNs per glomerulus are $N_{PN} = 5$ ans $N_{LN} = 3$ in qualitative agreement with what is reported in the literature [84, 89–91]. LNs are inhibitory whereas PNs are excitatory. For simplicity, we do not model the multiple kinds of LNs or PNs that have been observed in the AL. Similar models are being used extensively in the analysis of the insect AL [36, 37, 74, 87] and are well suited for replicating the competition dynamics that we seek to evaluate.

We model neurons as leaky integrate-and-fire (LIF) neurons with conductance based synapses. For PN $i$,

$$
\begin{aligned}
C\dot{V}_i &= g_l^{\text{PN}}(V_{\text{rest}} - V_i) + g_{\text{ORN}} s_i (V_{\text{rev}}^E - V) + g_{ad} x_i (V_{\text{rev}}^I - V_i) + g_{\text{LN}} y_i (V^I - V_i) + I_{\text{noise}}^{\text{PN}} \eta \\
\dot{x}_i &= \alpha_{\text{ad}} (1 - x_i) \sum_{\{t_{\text{spike}}^i\}} \delta(t - t_{\text{spike}}^i) - \frac{x_i}{\tau_{\text{ad}}}.
\end{aligned}
\tag{6}
$$

where $V_i$ is the membrane potential of PN $i$, $C = 1$ nF, $V_{\text{rev}}^E$ and $V_{\text{rev}}^I$ are reversal potentials for the excitatory and the inhibitory input, respectively. $x_i$ is the activation of a spike rate adaptation current with decay constant $\tau_{\text{ad}}$, $s_i$ is the combined activation of synaptic inputs from connected ORNs, $y_i$ is the activation of inhibitory synapses from connected LNs and $\eta$ Gaussian white noise with zero mean and standard deviation one. The LNs are described by a similar model but without adaptation,

$$C\dot{V}_i = g_l^{\text{LN}}(V_{\text{rest}} - V_i) + g_{\text{PN}} z_i (V_{\text{rev}}^E - V_i) + I_{\text{noise}}^{\text{LN}} \eta \tag{7}$$

where $V_i$ is the membrane potential of LN $i$, and $z$ represents the activation of excitatory synapses from the PN. For both PNs and LNs, when $V > \Theta$, the neuron fires a spike and $V$ is reset to $V_{\text{rest}}$ and it does not change for a refractory period $\tau_{\text{ref}} = 2$ ms.

Synaptic activation is modeled according to

$$\dot{\hat{s}}_j = \alpha_{\text{ORN}} (1 - \hat{s}_j) \sum_{\left\{ t_{\text{spike}}^{\text{ORN}_j} \right\}} \delta\left(t - t_{\text{spike}}^{\text{ORN}_j}\right) - \frac{\hat{s}_j}{\tau_{\text{ORN}}} \tag{8}$$

$$s_i = \sum_{j=1}^{n_{\text{ORN}}} w_{ij} \hat{s}_j \tag{9}$$

where $\left\{ t_{\text{spike}}^{\text{ORN}_j} \right\}$ are the times of all spikes in ORN $j$, and $w_{ij}$ is the connectivity matrix between ORNs and PNs, $w_{ij} = 1$ if ORN $j$ is connected to PN $i$ and $w_{ij} = 0$ otherwise. Similarly, for PN

to LN excitation,

$$\dot{\hat{z}}_j = \alpha_{\mathrm{PN}}\left(1 - \hat{z}_j\right) \sum_{\left\{\substack{\mathrm{PN}_j \\ t_{\mathrm{spike}}^{\mathrm{PN}_j}}\right\}} \delta\left(t - t_{\mathrm{spike}}^{\mathrm{PN}_j}\right) - \frac{\hat{z}_j}{\tau_{\mathrm{PN}}} \tag{10}$$

$$z_i = \sum_{j=1}^{n_{\mathrm{PN}}} v_{ij}\hat{z}_j, \tag{11}$$

with connectivity $v_{ij}$, and for LN to PN inhibition,

$$\dot{\hat{y}}_j = \alpha_{\mathrm{LN}}\left(1 - \hat{y}_j\right) \sum_{\left\{\substack{\mathrm{LN}_j \\ t_{\mathrm{spike}}^{\mathrm{LN}_j}}\right\}} \delta\left(t - t_{\mathrm{spike}}^{\mathrm{LN}_j}\right) - \frac{\hat{y}_j}{\tau_{\mathrm{LN}}} \tag{12}$$

$$y_i = \sum_{j=1}^{n_{\mathrm{LN}}} u_{ij}\hat{y}_j, \tag{13}$$

with connectivity $u_{ij}$. Each one of these variables has its decay time constant—$\tau_{\mathrm{ORN}}$, $\tau_{\mathrm{PN}}$, and $\tau_{\mathrm{LN}}$. The multiplicative factors $\alpha_{LN}$, $\alpha_{ORN}$ and $\alpha_{PN}$, reflect the amount of released vesicles for each pre-synaptic spike from an ORN and LN, respectively.

All parameters used for the simulations are reported in Table 1. The comparative analysis between LN inhibition and the NSI mechanism has been carried out through the exploration of the two parameters $\alpha_{LN}$ and $\omega_{\mathrm{NSI}}$.

## Simulation of realistic plumes

In a realistic scenario, odorants are mixed together in complex plumes that follow the laws of fluid dynamics (see e.g. [92]). For these conditions, even odorants coming from different sources are sometimes mixed together, and one difficult task for insects is to recognize when two intermingled odorants are coming from the same source or from separate sources. Of course, it is not possible to distinguish these two possibilities from a single, instantaneous sampling, but on average the odorants coming from the same source are more correlated than odorants coming from separate sources (see S5 Fig). To test the function of NSIs for odor source separation, we adopted long stimuli ($> 3s$), with statistical properties that resemble the filaments observed downwind from an odor source in an open environment ([1–3, 29] at zero crosswind distance. For these conditions, the distributions of whiff and clean air durations follow a power law with exponent -3/2 (see, e.g., [1]), and the cumulative distribution function (CDF) for the normalized concentration values will follow an exponential distribution which we fitted as a piece-wise function as follows

$$\begin{cases} 5/3x & 0 \leq x \leq 0.3 \\ 1 - 10^{-(a_1 + b_1 x)} & \text{otherwise} \end{cases}$$

where $x$ is the normalized concentration $C/\langle C \rangle$ and $a_1 = 0.22$ and $b_1 = 0.26$ are free parameters, whose values were determined by fitting the data reported in [3].

We analyzed stimuli with different maximum value of the whiff duration from $0.03s$ to $50s$. To simulate the arrival of plumes of two odorants with the aforementioned properties, we generated a time series of whiffs and blanks with the correct statistics for each odorant and the correct correlation between odorants. This was achieved with the following procedure:

1. We drew two correlated pseudo random numbers from a Gaussian distribution, with a given correlation

2. We mapped the two numbers into two uniform random variables

3. The uniform random variables are mapped into the desired power law distributions

The final result are time series for whiff-durations and -concentrations where for each odor the durations and concentrations are independently sampled from the desired distributions and we have approximately the prescribed correlation between the durations and concentrations between the time series of the two odors. We measured and report against the post-hoc value of correlation between the generated plumes.

The simulations were run with custom made Python code available online at https://github.com/mariopan/flynose.

## Analysis

**Firing rates.**   We analyzed the response of the model on the basis of the firing rates of the ORNs and PNs. The firing rate is calculated as a spike density function as reported in [65]. We calculated the convolution of spike trains using the asymmetric function $k(\hat{t}) = \hat{t} \exp(\hat{t}/\tau)$, where $\hat{t} = t - t_{spike} + \tau$ for the spike time $t_{spike}$. The timescale chosen was $\tau = 20$ ms.

**Maximum, peak, and average activity.**   We maintained an agnostic position regarding the way higher brain regions elaborate the signal coming from the AL. Although we are aware of the existence of more complex options, for simplicity we used three measures for the activity of the PNs: the *average*, *maximum* and the *peak activity*. The average activity is measured as the average rate of the neurons of the same glomerulus over 200ms after the stimulus onset. The maximum activity is measured as the maximum rate of each neuron within 200 ms after stimulus onset, averaged across all neurons of the same glomerulus. For the longer, more realistic odorant plumes, we used the definition of peak activity as the integrated activity over time windows where the neuron firing rate is above a given threshold (e.g. 50, 100, or 150 Hz).

**Encoding ratio.**   To test that NSI can help encode the odorant concentration ratio, we analyzed the ratio of PNs activity, $R_{PN}$, as follows: The PN activity is measured as the maximum activity. The encoding error, $err_{enc}$, is defined as squared relative distance from the concentration ratio, $R_{conc}$, that is:

$$err_{enc} = \left(\frac{R_{PN} - R_{conc}}{R_{PN} + R_{conc}}\right)^2 \tag{14}$$

In a similar way, we measure $err_{enc}$ for the *average activity* and the results are qualitatively identical.

## Supporting information

**S1 Fig. Model ORN response to a single step, a ramp, and a parabola as in [30].** Model ORN response to a single step (a,b), ramp (c,d), and parabola (e,f). a, c, e: Stimulus waveforms, i.e. odorant concentration profiles, as in [38]. b, d, f: Model ORN firing rates visualized as a spike density function (SDF).
(EPS)

**S2 Fig. Output of the model of Lazar and Yeh [30] for the Or59b receptor neuron in response to the corresponding stimulus waveforms (experimental data reported in [38]).**
(EPS)

**S3 Fig. PNs, ORNs and LNs response to a short (50 ms) pulse, similar results of Fig 4.** a) 50 ms step stimuli, shade of green indicates concentration. b)-d) corresponding activity of ORNs, PNs, and LNs. Shades of green match the input concentrations. e) Average response of PNs over 50 ms against the average activity of the corresponding ORNs. The orange dashed line is the fit of the simulated data using equation Eq 1 as reported in [41]. f) Average values for PNs, ORNs, and LNs for different values of concentration. Error bars show the SE over PNs. (EPS)

**S4 Fig. PNs, ORNs and LNs response to a short (100 ms) pulse, similar results of Fig 4.** a) 100 ms step stimuli, shade of green indicates concentration. b)-d) corresponding activity of ORNs, PNs, and LNs. Shades of green match the input concentrations. e) Average response of PNs over 100 ms against the average activity of the corresponding ORNs. The orange dashed line is the fit of the simulated data using equation Eq 1 as reported in [41]. f) Average values for PNs, ORNs, and LNs for different values of concentration. Error bars show the SE over PNs. (EPS)

**S5 Fig. Plume statistics of natural plumes.** Whiff and blank durations are defined as the time period of non-zero and zero odor concentration, respectively, as measured at a stationary point downwind from a source. Whiff durations can depend on wind speed and can differ for flying insects depending on flight speed and direction relative to odor filaments. a) Probability distribution of the whiff durations for odorants emitted at distances larger than 60 m [2]. b) Probability distribution of the blank durations for odorants emitted at distances larger than 60 m [2]. Wind speeds in a) and b) were between 1 ad 5 m/s [2]. c) Probability distribution of the normalized concentration for odorants emitted at 75 m distance from the source [3]. (EPS)

**S6 Fig. Firing rates of ORNs (as maximum activity) in response to a single synchronous triangular pulse of 50 ms duration applied to both ORN groups.** The activities are calculated for several values of odor concentration ratio of the two odorants and for six different overall concentrations. (EPS)

**S7 Fig. Firing rates of PNs (as maximum activity) in response to a single synchronous triangular pulse of 50 ms duration applied to both ORN groups.** The activities are calculated for several values of odor concentration ratio of the two odorants and for six different overall concentrations. (EPS)

**S8 Fig. Encoding ratio with the average PN activity.** ORN (a, c, e, g) and PN (b, d, f, h) responses to a single synchronous triangular pulse of 50 ms duration applied to both ORN groups. The graphs show average responses ratio ($R^{ORN}$ and $R^{PN}$), respectively, versus concentration ratio of the two odorants for six different overall concentrations (colours, see legend in d). The average PN responses would be a perfect reflection of the odorant concentration if they followed the black dashed diagonal for all concentrations. Error bars represent the semi inter-quartile range calculated over 50 trials. i) Analysis of the coding error for different values of stimulus duration and concentration values. (EPS)

**S9 Fig. Correlation analysis between the odorant concentration ratio and PNs response ratio, $R^{PN}$, for different values of stimulus duration (from 10 to 200ms) and concentration**

values (reported values refer to the absolute values of the odorant with lower concentration).
(EPS)

**S10 Fig. Example concentration fluctuation time series of natural plumes for two odorants emitted by a single source or two separate sources [16].**
(EPS)

**S11 Fig. Comparison of the median ratio of PN responses under asynchronous pulses for different stimulus durations, delays and for two values of the synaptic decay time constant ($\tau_{LN}$).** Panel a shows the results using $\tau_{LN} = 250ms$ and panel b $\tau_{LN} = 25ms$. The median ratio of the average PN responses of the two glomeruli is defined as $R^{PN} = v_b^{PN}/v_a^{PN}$. The line styles and colors indicate the four models: control model (pink), LN model (orange), mix model (green) and NSI model (cyan). The stimulus durations are marked on the top. Error bars represent the semi inter-quartile ranges.
(EPS)

**S12 Fig. Responses to asynchronous pulses of the control, LN, NSI and mix model, similar to Fig 8 but for a short synaptic time scale $\tau_{LN} = 25ms$.** Panel a) Time course of ORN and PN activity in response to two asynchronous triangular pulses (50 ms, 1st column) for the four models—control (pink), NSI (cyan), mix (green) and LN model (orange). Input to the four models is identical, while control and LN models have identical ORN activity—as much as mix and LN—which is therefore only displayed once. The continuous and dashed lines represent the two odors, ORN and PN types. The lines show the average response over 10 trials. Panel b) Median ratio of the maximum PN responses of the two glomeruli $R^{PN} = v_b^{PN}/v_a^{PN}$ in the four models: control model (pink), LN model (orange), mix model (green) and NSI model (cyan) for stimulus duration of 50 ms as marked on the top. Error bars represent the semi inter-quartile ranges.
(EPS)

**S13 Fig. Statistical properties of simulated natural plumes.** Observed properties of the simulated plumes as a function of the intended correlation between plumes averaged over 200 s. Intermittency and average input plots show the values for the two plumes (green and purple).
(EPS)

**S14 Fig. Total PN activity above 50, 100, 150 Hz, respectively, for 3 s maximum whiff durations, similar results of Fig 9d.**
(EPS)

**S15 Fig. Comparing encoding efficiency of odorants' correlation, similar results of Fig 10 using threshold 50 Hz.** a) Peak PN response for threshold 50 Hz and for different subsets of maximal whiff durations (from 0.03 to 50s) for the four models: control model (pink), LN model (orange), mix model (green) and NSI model (cyan). b) Distance between the PN activity of the control model and one of the other three models (NSI, LN or mix model), at 0 correlation, $p_{ctrl}^0 - p_x^0$ with $x \in$ (NSI, LN, mix). c) Distance between the PN activity at 0 correlation and at correlation 1 for the NSI, LN and mix model, i.e., $p_x^0 - p_x^1$ with $x \in$ (NSI, LN, mix). Note that the horizontal axis has a log-scale.
(EPS)

**S16 Fig. Comparing encoding efficiency of odorants' correlation, similar results of Fig 10 using threshold 150 Hz.** a) Peak PN response for threshold 150 Hz and for different subsets of maximal whiff durations (from 0.03 to 50s) for the four models: control model (pink), LN model (orange), mix model (green) and NSI model (cyan). b) Distance between the PN activity

of the control model and one of the other three models (NSI, LN or mix model), at 0 correlation, $p^0_{ctrl} - p^0_x$ with $x \in$ (NSI, LN, mix). c) Distance between the PN activity at 0 correlation and at correlation 1 for the NSI, LN and mix model, i.e., $p^0_x - p^1_x$ with $x \in$ (NSI, LN, mix). Note that the horizontal axis has a log-scale.
(EPS)

**S17 Fig. Low and high threshold (panels a and b, respectively) of the two models calculated for several values of the ORNs sensitivity distance ($\log_{10}(C^b_b/C^a_l)$) in order of magnitude (OM).** For the control (pink) and the NSI model (cyan), dynamic range, low and high threshold are calculated from the average activity of the two ORNs. For comparison, see the lower and higher thresholds one of the isolated ORNs (green).
(EPS)

## Author Contributions

**Conceptualization:** Mario Pannunzi, Thomas Nowotny.

**Data curation:** Mario Pannunzi.

**Formal analysis:** Mario Pannunzi, Thomas Nowotny.

**Funding acquisition:** Mario Pannunzi, Thomas Nowotny.

**Investigation:** Mario Pannunzi, Thomas Nowotny.

**Methodology:** Mario Pannunzi, Thomas Nowotny.

**Project administration:** Mario Pannunzi, Thomas Nowotny.

**Resources:** Mario Pannunzi, Thomas Nowotny.

**Software:** Mario Pannunzi, Thomas Nowotny.

**Supervision:** Thomas Nowotny.

**Validation:** Mario Pannunzi, Thomas Nowotny.

**Visualization:** Mario Pannunzi, Thomas Nowotny.

**Writing – original draft:** Mario Pannunzi, Thomas Nowotny.

**Writing – review & editing:** Mario Pannunzi, Thomas Nowotny.

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
