## [Decision Letter · Decision Letter 0]

2 Aug 2021

Dear Dr Pannunzi,

Thank you very much for submitting your manuscript "Non-synaptic interactions between olfactory receptor neurons, a possible key feature of odor processing in flies" for consideration at PLOS Computational Biology.

As with all papers reviewed by the journal, your manuscript was reviewed by members of the editorial board and by several independent reviewers. In light of the reviews (below this email), we would like to invite the resubmission of a significantly-revised version that takes into account the reviewers' comments.

We cannot make any decision about publication until we have seen the revised manuscript and your response to the reviewers' comments. Your revised manuscript is also likely to be sent to reviewers for further evaluation.

Sincerely,

Michele Migliore

Associate Editor

PLOS Computational Biology

Thomas Serre

Deputy Editor

PLOS Computational Biology

Reviewer's Responses to Questions

**Comments to the Authors:**

Reviewer #1: uploaded as an attachment

Reviewer #2: Insect olfaction is intensely studied and computationally challenging. The olfactory system, both of invertebrates and vertebrates, performs intricate computations that transform a high-dimensional and temporally highly dynamics input space into a meaningful spatio-temporal code that underlies animal perception and behaviour. The present MS deals with an important but hitherto mostly neglected feature of the experimentally demonstrated non-synaptic interaction (NSI) among sensory neurons at the earliest peripheral stage. In their computational model study, the authors compare the effect of peripheral NSI to the well-studied model of lateral inhibition in the antennal lobe and conclude that NSI is beneficial for representing the ratio of two odours in a binary mixture and for detecting temporally asynchronous mixtures.

While I believe that the topic and the computational approach of the present MS is very well suited for PLoS CB, my numbered comments below raise several questions and concerns that need to be addressed in the course of a manuscript revision. In particular, the description of model and methods is in many parts incomplete or missing (!), which complicated the review. Also, my concerns with respect to (i) the ORN adaptation time constant (and the resulting effect of ORN adaptation) and (ii) the very long inhibitory synaptic time constants need to be addressed as I believe they are crucial for correct interpretation of model comparisons and results.

MODEL & METHODS

1) Missing descriptions in the “model and methods” section

Unfortunately, in the submitted MS version parts of the method section seem to be missing. The paper repeatedly refers to the methods section for detailed explanations but I could not find any of the expected explanations. This makes it hard and often impossible to fully understand and review the MS. For example:

- Line 240 mentions the coding error without clear definition (measured over time course? For peak responses?) and caption of Fig. 6 refers to the method section, but nothing is there. It is also mentioned as either squared relative distance or relative distance at different instances of the text body.

- how is firing rate estimated?

- ratio of peak activity?

- Line 307 “The statistics of the plumes and how we simulated them are described in detail in the Model and methods; …” – no, nothing there. A detailed description is particularly relevant if others would like to use this input model for comparison.

- Line 603 “In addition to the inputs from ORNs, PNs also receive global excitation from PNs associated with other glomeruli and from other parts of the brain.” I could not identify where and how this global excitation enters the model. The noise in equation (6) has zero mean and there is no other excitatory input specified. Does equation (6) lack a term for constant current input or similar?

2) Model parameters

Table 1 – It would be helpful to indicate e.g. by bold font which of the parameters have been fitted to experimental data and which have not. By what method/approach was the fitting of all 24 parameters performed?

The LIF ORN is defined as voltage equation and parameters are provided as dimensionless numbers except for resting, reversal and threshold voltages. This choice makes it difficult to interpret these parameters relative to the parameters of the other neurons. What is the dimension of the adaptation time constant? Please clarify.

Please carefully check nomenclature in 4.2. E.g. a_y, b_y, b_r versus alpha_y , beta_y, beta_r.

Which parameter is meant with d_r in line 621: “The parameters ay and by are estimated together with br and dr to reproduce the data presented in [28].” – does this refer to the parameter alpha_r?

Capacity “C” for PN and LN is not given.

Please carefully check for other parameters that might not be defined such as e.g. concentration.

RESULTS

3) Concentration ratio

Fig. 6f shows one of the central results that is very interesting. The ratios of response rates show a clear picture. What are the absolute rates, e.g. of the dominant odor? Are these still in a realistic range?

I do not understand Fig. 6g: “It is clear that NSI model outperforms both LN and control

model, apart for very high concentrations.” In Fig. 6a-f the authors look at the concentration ratio of the two odorants without providing absolute concentrations for any of the two odours. What is the absolute concentration in a-f? In Fig. 6g. only an absolute concentration is provided on the y-axes for the 2d calibration plot. Which concentration ratio was chosen for panel g? This is puzzling, please clarify and use the caption to provide full information.

4) Correlation analysis

The analysis in Fig. 8 is difficult to assess due to the missing explanation of the plume statistics. The caption reads “… response … to two 4 s long realistic plumes with

statistical properties as described in the text;” but I cannot find this description. For panel d it says that the max. whiff duration equals 3ms. This is on the very short end of the statistics, does this hold for all results in Fig. 8?

5) Adaptation

The chosen model of spike frequency adaptation in the ORNs introduces a mode of self-inhibition. Table 1 indicates an ORN adaptation time constant of 0.0035 - is this to be interpreted as 3.5ms – please clarify? How does this number fit to time constants reported in the experimental and recent model literature that investigated ORN adaptation (e.g. ref 28 and others not referenced)? A time constant of ~3ms would seem extremely short (e.g. compared to 258ms in the PN) and I would expect to see no dynamic effect of adaptation in the resulting response profile because the adaptation current effectively decays over less than one ISI. However, the firing rate dynamics in Fig. 3 looks realistic to me, both in amplitude and temporal evolution. I would have expected that the phasic-tonic responses to a step stimulus are due to the adaptation. However, a decay time constant of 3.5ms and a maximum rate of 100Hz corresponding to an average 10ms inter-spike interval indicates that the adaptation profile of the firing rate (adapting over the course of several hundreds of ms) is not due to the spike frequency adaptation. Can the authors explain mechanistically how this response rate adaptation is brought about? If it is due to the ORN adaptation current then I would like to see the response in Fig. 3 when adaptation is switched off for comparison (possibly with compensating average negative charge effect).

L 544 the authors argue that adaptation is an important element of odor processing. If the adaptation time constant in the ORN spike generator has really been set to only 3.5ms then I doubt that it is of relevance in the authors’ ORN model.

The assumingly short ORN adaptation time constant might have little effect on asynchronous mixture encoding. A longer realistic time constant of tens to hundreds of ms (similar to the very long LN inhibitory synaptic time constant) might have?

6) Lateral inhibition

The synaptic time constant tau_LN is set to 250ms as a standard (compared to 19ms for tau_PN) and thus is of the same duration as the PN adaptation time constant (tau_ad=258ms). Is that a realistic number? To me synaptic time constants of 20ms seem already long, 250ms seems very specific. Where the synaptic time constants among the fitted parameters or were they chosen freely? Are there numbers in the literature that can support these long time constants or is this a model prediction? Since LNs are non-adapting, could the LN->PN inhibition have a similar effect as the PN adaptation, i.e. an outward conductance that can sum up over the stimulus period with similar long time constants as the PN adaptation?

In line 303 the authors state that the NSI “… distort responses less than LN inhibition in the case of asynchronous mixtures.” However, as they state this is not the case for ‘short’ inhibitory synaptic time constants of 25ms (which to me still seems a very long synaptic time constant). Thus, this seeming disadvantage of the LN model depends on the choice of time constant. This needs to be made explicit here and in the Discussion.

7) PN response saturation

The result in Fig. 4E is nice and fits well the experimental observations. Which mechanism causes the saturation effect in the PN firing rate with respect to ORN input rate? The authors should explain the causes of this effect.

8) Above threshold rate encoding

The authors state that “A biologically plausible and commonly adopted hypothesis argues that information is conveyed in the high activity, i.e. firing rates above a given threshold.” The authors are asked to provide references here. I find this approach rather critical. Results in Fig. 8 and supplements seem to depend critically on this chosen threshold. Given the avg. PN rates of 30 – 45 Hz in Fig. 8c and multiplying those by 5 PNs the total activity would be around 150 – 225Hz. The 150Hz threshold criterion thus implies that most of the PN activity as base activity is neglected.

What does “Total PN activity above 150 Hz.” (caption Fig. 8) actually mean? Total as the sum across all 5 PNs per glomerulus? What does a number of 4.000 on the y-axes of Fig. 8d mean? Is this an integrated measure as indicated by line 449 in the Discussion? Why are units of this measure arbitrary, should this be total number of spikes as in time-integrated rate or completely meaningless? Please specify the definition of this threshold model in the text and method section. Again, the method section does not describe this approach at all.

In light of this unclear definition I find it hard to interpret line 452f “… this also adds further evidence in favor of using peak activity to encode important features of a signal, in this case stimulus correlations, as hypothesized in earlier work (see e.g. [19, 52]).” What exactly do the authors mean here with “peak activity” with respect to their analyses: the peak of a time-varying rate above some high threshold? The integrated rate above some threshold? And what definition of peak activity has been used in the two references? Are these definitions coherent?

DISCUSSION

9) As the authors state, the model does not capture the underlying physics of the NSI mechanism (line 480 f). While implementing the underlying mechanism into the model is clearly out of the scope of this study, it would however be interesting to shortly review what is hypothesized to be relevant mechanisms behind NSIs and/or discuss plausible mechanisms.

10) A number of studies have shown that both, NSIs and LI, are key features in the olfactory processing in the fly. In the conclusion of the manuscript, the authors shortly mention that it would be interesting to study their differential roles when both mechanisms are present (line 593 f.). In my opinion, joint and mutual effects of NSIs and LI should be discussed in more detail. How do the two mechanisms possibly complement each other (e.g. narrow and specific connectivity between two ORNs in sensilla and broad connectivity/inhibition through LNs in AL)?

MINOR

Fig.5: input in b, left panel is dimensionless? The standard deviation over 10 trials is not visible and essentially zero; thus I assume that the displayed rate is in each trial averaged across all 80 ORNs and 5 PNs in the right panel? How was firing rate estimated, this is not indicated in the methods?

L 224: “Fig.6 based on the ratio of peak activity R = nu_b=nu_a (both for ORNs and PNs, see Model and methods) during the first 200 ms after the stimulus onset.” Unfortunately the methods section does NOT explain the stimulus protocol or response estimates of nu_a, nu_b. It would be lepful to make clear here that this is related to the peak firing rate responses and again the method for rate estimation should be defined in the method section.

Correlation analysis in Fig. 8: I understand that the correlation among odor inputs is simulated and controlled by some model parameter (model not described, see 1 above). Why does the x-axes then read “Observed correlation”? Is there any additional step of correlation analysis that leads to these x values? Please clarify.

L 355: w_max = 50m vs. 50s ?

L 372: The subscripts should read “NSI” or “LN” (and not “x”) in both subtraction terms, respectively.

Fig. 5a is only mentioned in the text in line 195 ff. (“The whiffs in plumes (see e.g. Fig.S5a) are mimicked with simple triangular odorant concentration profiles that have a symmetric linear increase and decrease (see Fig.5a).” I think this statement refers to Fig. 5b and apart from that, Fig. 5a is not mentioned in the text.

Fig. 5b: The difference between the dashed NSI line) and the dot-dashed LN line is not very clear (applies even more to PN panel in Fig. 7a). Here, color coding / more different line styles would increase discriminability.

The Figure referencing in inconsistent (e.g. “see Fig.1b” in line 54; “see Fig.1, panels c-d” in line 56; “8a-b” in line 329),

Some subfigures are referenced wrongly (line 197: Fig.5b instead of Fig.5a; caption of Figure 6: “two odorants for four different overall concentrations (colours, see legend in f)” should be “six different concentrations” and “see legend in d”)

Caption for Fig 10c is missing,

Check line 355: “from 0.01 to 50 m” should be “0.03” and “s” ?

**Have the authors made all data and (if applicable) computational code underlying the findings in their manuscript fully available?**

Reviewer #1: Yes

Reviewer #2: Yes

PLOS authors have the option to publish the peer review history of their article (what does this mean?). If published, this will include your full peer review and any attached files.

Reviewer #1: No

Reviewer #2: **Yes: **Martin Nawrot
---

## [Decision Letter · Decision Letter 1]

22 Oct 2021

Dear Dr Pannunzi,

We are pleased to inform you that your manuscript 'Non-synaptic interactions between olfactory receptor neurons, a possible key feature of odor processing in flies' has been provisionally accepted for publication in PLOS Computational Biology.

Best regards,

Michele Migliore

Associate Editor

PLOS Computational Biology

Thomas Serre

Deputy Editor

PLOS Computational Biology

Reviewer's Responses to Questions

**Comments to the Authors:**

Reviewer #1: The authors have addressed all the comments in the revised version.

Reviewer #2: The authors have made substantial changes and amendments that have clearly improved the manuscript in all parts. The missing subsection in the Methods section have been added and method descriptions and tables have been revised resulting in more clarity. The results have been augmented by supplemental material that provides additional transparency, e.g. with regard to the absolute firing rates of ORNs and PNs that vary within a realistic range. I highly welcome the addition of the mixed model of NSI and lateral inhibition that reviewer 1 has been asking for and that I had been suggesting when reviewing an earlier version of the MS. This addition has clearly advanced the manuscript.

As stated in my original review, I believe that this is a really interesting manuscript and the computational and conceptual treatment of NSIs in (inset) olfaction is truly novel. I thus recommend acceptation of the manuscript for publication with PLoS CB.

Below I have a few final constructive comments/questions that I would like to provide to the authors that can/should be used for final improvements before publication. In parts they refer to the previously absent methods part asking for some more clarity there. Others refer to specific figure panels.

#1 Methods subsection 4.5: Simulation of realistic plumes

How is “whiff duration” defined? As I understand it the definition is given by the empirical data used, which is referred to as “filaments observed downwind”. From this I infer that the authors assume the presence of a “downwind” and whiff duration at a stationary point in space is then likely brought about by wind rather than by diffusion or turbulences in a wind-free scenario. No animal movement such as in a flying insect is considered. Is this correct? Please state this explicitly and if possible, provide the estimate of wind speed for the empirical data if that exists. Adding flight speed might considerably shorten the encountered whiff durations, in walking animals this would be negligible.

I am not sure whether I fully understand the approach of sequence generation (without checking the online code): “Whiff duration” is drawn from a power law distribution with exponent -3/2. This distribution is truncated at different maximum values between 0.03s to 50s. Why these different truncation values? The duration of odor stimuli was “>3s”, so I assume that the truncation of the distribution has to do with the duration of the stimulations? Please specify.

For the concentration the authors assume an exponential CDF. Are the concentrations per whiff drawn from this CDF (and if so, independently for both odours?) or is the concentration fix across all whiffs for a given scenario? The authors probably assume a fix given distance from the source. What would that distance be? Or do the different truncations of the power law distributions actually reflect different distances from the source?

Now, I still cannot easily follow the 3-step procedure outlined in lines 870 – 873, please make this more explicit: The authors drew correlated random numbers from a Gaussian distribution, these are mapped to uniform random variables and used to index into power law distributions. The power law distributions would be two, namely the distributions of whiff durations and blank durations. I assume the same durations for whiff and blank durations where used for the two different odors? If correlation is 1, then the very same durations apply for both odours and the sequences of alternating whiffs and blanks are then concatenated to generate the very same sequence of whiffs and blanks for both odors, is this correct? If the correlation is 0<c<1, a="" actual="" and="" are="" be="" but="" correlated="" correlation="" decay="" description.="" different="" diverge="" generated="" i="" in="" little="" maybe="" misinterpreted="" my="" odour="" rapidly="" sequences="" series="" the="" then="" there="" time="" time.="" to="" two="" understanding="" very="" way="" will="" with="" would="">

#2 Fig. 7 f,h (former Fig. 6)

I compare Fig. 7f with former Fig. 6f, these are the same panels, meaning they describe the ratio of projection neuron firing rate as a function of the concentration ration in the NSI model. Now the results in the revised MS are quite different from the previous result where the NSI model seemed to have performed better (in the sense of better achieving the desired identity mapping). In the revised MS panel 7h shows the performance of the mixed model, which comes close the previous performance of the NSI model.

I assume the reason for this is that, to accommodate the mix model, the authors have modified and newly tuned the NSI model. I can see in Table 1 that some the parameter values have changed. However, I did not find any reference to this in the reply to the comment #1 of reviewer 1 or to my comments on the discussion (or I have missed it). While I think it is fair to further improve and newly tune the model for MS revision I want the authors to double check that the right data is shown in the respective panels (in particular in 7f).

#3 Fig. 9 (former Fig. 8):

The y-axis on panel d is still “unitless” but from the response to my question the authors described that peak activity is peak rate, so the units should be Hz or ½, shouldn’t it?</c<1,>

**Have the authors made all data and (if applicable) computational code underlying the findings in their manuscript fully available?**

Reviewer #1: Yes

Reviewer #2: Yes

PLOS authors have the option to publish the peer review history of their article (what does this mean?). If published, this will include your full peer review and any attached files.

Reviewer #1: No

Reviewer #2: **Yes: **Martin Paul Nawrot

---

## [Editor Report · Acceptance letter]

15 Nov 2021

PCOMPBIOL-D-21-01196R1 

Non-synaptic interactions between olfactory receptor neurons, a possible key feature of odor processing in flies

Dear Dr Pannunzi,

I am pleased to inform you that your manuscript has been formally accepted for publication in PLOS Computational Biology. Your manuscript is now with our production department and you will be notified of the publication date in due course.

With kind regards,

Livia Horvath
